# Photoactivatable Cre recombinase 3.0 for in vivo mouse applications

Kumi Morikawa[1,2,16], Kazuhiro Furuhashi[1,3], Carmen de Sena-Tomas [1,4], Alvaro L. Garcia-Garcia[5],
Ramsey Bekdash[1,2,6], Alison D. Klein[1,2], Nicholas Gallerani[1,7], Hannah E. Yamamoto[1,2,8], Seon-Hye E. Park[1,2,9],
Grant S. Collins[10], Fuun Kawano[1,2,11], Moritoshi Sato [11,12], Chyuan-Sheng Lin[7,13], Kimara L. Targoff[1,4],
Edmund Au[1,7,14], Michael C. Salling[10,15] & Masayuki Yazawa[1,2,6,12 ✉]

Optogenetic genome engineering tools enable spatiotemporal control of gene expression and provide new insight into biological function. Here, we report the new version of genetically encoded photoactivatable (PA) Cre recombinase, PA-Cre 3.0. To improve PA-Cre technology, we compare light-dimerization tools and optimize for mammalian expression using a CAG promoter, Magnets, and 2A self-cleaving peptide. To prevent background recombination caused by the high sequence similarity in the dimerization domains, we modify the codons for mouse gene targeting and viral production. Overall, these modifications significantly reduce dark leak activity and improve blue-light induction developing our new version, PA-Cre 3.0. As a resource, we have generated and validated AAV-PA-Cre 3.0 as well as two mouse lines that can conditionally express PA-Cre 3.0. Together these new tools will facilitate further biological and biomedical research.

[1] Columbia Stem Cell Initiative, Columbia University, New York, NY 10032, USA. [2] Department of Rehabilitation and Regenerative Medicine, Vagelos College of Physicians and Surgeons, Columbia University, New York, NY 10032, USA. [3] Columbia Center for Translational Immunology, Vagelos College of Physicians and Surgeons, Columbia University, New York, NY 10032, USA. [4] Department of Pediatrics, Vagelos College of Physicians and Surgeons, Columbia University, New York, NY 10032, USA. [5] Department of Psychiatry, Division of Systems Neuroscience, New York State Psychiatric Institute, Columbia University, New York, NY 10032, USA. [6] Department of Pharmacology, Vagelos College of Physicians and Surgeons, Columbia University, New York, NY 10032, USA. [7] Department of Pathology and Cell Biology, Vagelos College of Physicians and Surgeons, Columbia University, New York, NY 10032, USA. [8] Barnard College, New York, NY 10027, USA. [9] Department of Neuroscience, University of Texas Southwestern Medical Center, Dallas, TX 75390-911, USA. [10] Department of Cell Biology and Anatomy, Louisiana State University Health Sciences Center, New Orleans, LA 70112, USA. [11] Graduate School of Arts and Sciences, The University of Tokyo, Tokyo, Japan. [12] Core Research for Evolutional Science and Technology, Japan Science and Technology Agency, Saitama, Japan. [13] Transgenic Mouse Shared Resource, Herbert Irving Comprehensive Cancer Center, Columbia University, New York, NY 10032, USA. [14] Columbia Translational Neuroscience Initiative Scholar, Columbia University Irving Medical Center, New York, NY 10032, USA. [15] Department of Anesthesiology, Vagelos College of Physicians and Surgeons, Columbia University, New York, NY 10032, USA. [16] Present address: Department of Life Science and Biotechnology, National Institute of Advanced Industrial Science and Technology, Tsukuba, Japan. ✉email: my2387@columbia.edu

The ability to manipulate DNA recombination and gene expression in a spatiotemporal specific manner is a powerful technique in genome engineering studies[1]. Cre recombinase, which is derived from P1 bacteriophage, is the most common recombinase that has been used to catalyze directional DNA recombination between *loxP* pairs[2–5]. While various inducible systems have been developed based on Cre-*loxP* recombination, the tools often suffer from either low efficiency (such as with the CRY2-CIB1-based system) or have complications such as the necessity for harmful chemical inducers such as tamoxifen or rapamycin[6–9]. While our previously reported Magnets-based PA-Cre system improved on many of these shortcomings[10], PA-Cre still had a major issue with unintentional recombination in dark conditions prior to light stimulation. In addition, there are currently no in vivo mouse models available for optogenetic-based systems, limiting the scope of applications in biological study.

In this study, we developed an improved version of PA-Cre called PA-Cre 3.0, which is based on the same blue-light-dependent dimerization system, Magnets. We demonstrate the improved efficiency of PA-Cre 3.0 and its applications in vivo using newly generated mouse lines expressing PA-Cre 3.0 conditionally. We believe this improved system and mouse model availability can enhance genetic studies in living systems to address biological hypotheses and unveil the molecular and pathophysiological mechanisms underlying various diseases.

## Results

**Improvement of PA-Cre system**. Previously, we developed the 1st generation of Magnets-based PA-Cre, taking advantage of blue-light-dependent hetero-dimerization system, Magnets[10]. Although the construct of PA-Cre could be successfully transiently applied in mammalian cells in vitro and mouse livers in vivo using hydrodynamic tail vein (HTV) injection and reporter plasmids, other blue-light-inducible hetero-dimerization systems could be more suitable for developing PA-Cre systems. To address this question, CRY2/CIB1-, iLID/SspB-, and FKF1/GI-based PA-Cre constructs were prepared and tested using luciferase (Luc) and mCherry reporters, and compared to the original Magnets-based PA-Cre[9,11–13] (Fig. 1a–c and Supplementary Fig. 1a, b). We found that the Magnets-based original PA-Cre had the highest Cre-*loxP* recombination efficiency with light among these constructs (~75% compared with a positive control, CreERT2, treated with tamoxifen). While the fold induction of Cre-*loxP* recombination using the CRY2/CIB1-based PA-Cre is the best (43.3×) among the tests, the efficiency of Cre-*loxP* recombination with light was low (~15%) compared with CreERT2 positive control. The FKF1/GI-based version also demonstrated as low Cre-*loxP* recombination efficiency with blue light as the CRY2/CIB1-based one (called PA-Cre 2.0)[11]. On the other hand, the iLID/SspB-based version had much higher leakiness in dark than the others. These results suggest that the original Magnets-based PA-Cre is still promising for further improvement as the unintentional Cre-*loxP* dark leak recombination is limited (Supplementary Fig. 1a). To assess this dark leakiness issue further, we monitored the Luc activity 24, 48, 72, and 96 h after HEK 293T cells were transfected with the PA-Cre constructs. The CRY2/CIB1-based construct, called PA-Cre 2.0 (ref. [11]), was also tested as a benchmark experiment. The Magnets-based PA-Cre showed an accumulating leak over time while the CRY2/CIB1 version demonstrated little to no leakiness (Supplementary Fig. 1c). Such leaky recombination in dark after PA-Cre expression is not acceptable for any in vivo applications as Cre-*loxP* recombination is irreversible. To address this issue, we looked to improve the Magnets-based PA-Cre system by reducing the background dark activity.

To characterize the original PA-Cre further, we conducted protein expression profiling using western blotting with the overexpression of PA-Cre construct in HEK 293T cells. Interestingly, we found that anti-Cre blotting demonstrated non-cleaved form of PA-Cre proteins at ~75 kDa although the most efficient self-cleaving peptide, P2A, has been utilized for this PA-Cre construct[10] (Supplementary Fig. 2a–c). Because non-cleaved PA-Cre proteins might not work properly, we hypothesized that the non-cleaved form could be associated with higher leakiness of Cre-*loxP* recombination with the original PA-Cre in the dark condition. To answer this hypothesis, we prepared a new PA-Cre construct using F2A, which is known to have lower efficiency of self-cleavage than P2A[14]. In addition, we substituted proline to glycine in the recognition site in order to disrupt P2A-mediated peptide self-cleavage completely to examine whether this mutation affects the leak activity of PA-Cre in dark (Supplementary Fig. 2a, b). Protein characterization using western blotting revealed that the mutated P2A and F2A constructs had significantly increased non-cleaved proteins at ~75 kDa compared with the original P2A-mediated PA-Cre construct in the dark condition (Supplementary Fig. 2c). The interrupted P2A cleavage and F2A version did in fact increase leakiness of Cre-*loxP* recombination in dark as confirmed via both Luc assay and mCherry reporter (Supplementary Fig. 2d–f).

Because nMag and pMag show 99% match in their DNA sequences, there is another concern that such a high similarity may increase the possibility of unintentional recombination in the mouse genome when PA-Cre is applied for transgenic mouse generation. To remedy this concern, we modified the codon of nMag together with CreN using DNA synthesis and *Nhe*I and *BamH*I cloning sites (i.e., Magnets-opti, Supplementary Fig. 3). After this codon modification, the DNA sequence of nMag in Magnets-opti has ~79% similarity with pMag without amino acid substitution (Fig. 2a). In addition, we tested not only the promoter of cytomegalovirus (CMV) but also CAG promoter, which contains the promoter of chicken beta-actin gene[15]. This is because the CMV promoter may be problematic for further in vivo applications as it has been shown to silence the promoter activity in murine systems in vivo while the CAG promoter has shown stronger activity than CMV without silencing issues[16,17]. To examine whether promoter choice affects the efficiency of P2A self-cleavage in the PA-Cre constructs, western blotting was conducted to profile each PA-Cre construct. We found the nMag codon-modified PA-Cre (Magnets-opti) had less non-cleaved form than the original ones while the total protein expression of Magnets-opti was also reduced (Fig. 2b). The CAG promoter-mediated Magnets-opti construct showed the best induction of Luc (382×) compared to the other iterations (Fig. 2c and Supplementary Fig. 4a). Next, we examined the IRES version of PA-Cre constructs to express CreN59-nMag-NLS and NLS-pMag-CreC60 separately. We found that the IRES was not useful for improving the PA-Cre system (Fig. 2c). This result was also confirmed using a mCherry reporter (Supplementary Fig. 4b–d), allowing us to develop our new version of PA-Cre construct using CAG promoter and nMag codon-modification called PA-Cre 3.0.

To validate the function of PA-Cre 3.0 in vivo, we induced transient expressions of the CAG promoter-driven Magnets-based constructs, both PA-Cre 3.0 and the original one, using HTV injection into mice. Following the plasmid DNA injection, mouse abdomens were exposed to blue light to induce recombination, following the experimental setup of our previous study in which the Luc reporter plasmids were also used with the PA-Cre construct[10]. However, the validation of Cre-*loxP* recombination for targeting a single pair of *loxP* sites is essential for applying the PA-Cre system in vivo. To address this, we utilized Ai14:Floxed-tdTomato heterozygous reporter mouse line, which can express tdTomato red fluorescent proteins in a

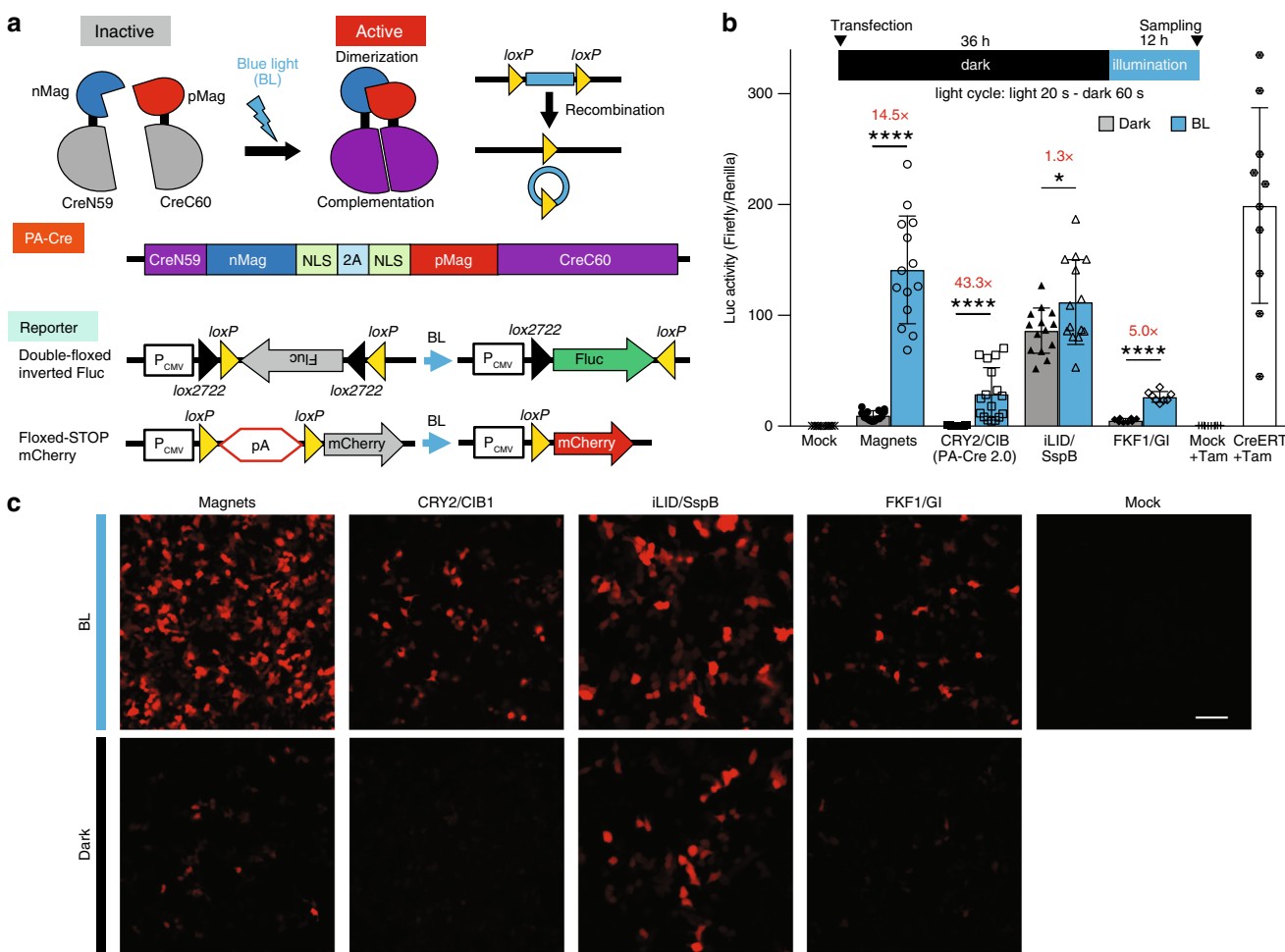

**Fig. 1 Comparison of multiple light-activated dimerization systems in photoactivatable Cre recombinase. a** Schematic representation of photoactivatable (PA)-Cre system and its reporter constructs. Split Cre (59/60) are complemented along with nMag–pMag dimerization upon blue-light illumination (BL blue light, NLS nuclear localization signal, 2A P2A self-cleaving peptide sequence, $P_{CMV}$ cytomegalovirus promoter, Fluc *Firefly* luciferase, pA polyadenylation transcriptional "stop" (poly-A) signal repeated sequence). **b** Comparison of PA-Cre with various blue-light photoreceptors using luciferase (Luc) assay. Upper diagram shows experimental protocol used for Luc assay (blue LED, 447.5 nm, 8.28 W/m², repeated 20 s light and 60 s dark for 12 h). Luc assays were conducted with double-floxed inverted Fluc reporter in HEK 293T cells. The herpes simplex virus thymedine kinase (HSV-TK) promoter-*Renilla* Luc plasmid was co-transfected as a transfection control to normalize *Firefly* Luc activity. CreERT2 vector and pcDNA3.1 empty vector (Mock) treated with 1 μM tamoxifen (Tam) addition 36 h after the transfection were used as positive and negative controls, respectively (*$P < 0.05$; ****$P < 0.0001$; dark v.s. BL using Two-tailed $t$ test, biologically independent samples; Mock, Magnets, iLID/SspB: $n = 14$, CRY2/CIB $n = 17$, FKF1/GI $n = 8$, Mock + Tam, CreERT2 + Tam: $n = 10$, mean ± s.d.). **c** Fluorescence images of HEK 293T cells expressing multiple versions of PA-Cre constructs and Floxed-mCherry reporter transiently ($n = 3$–7). Scale bar, 100 μm. Source data are provided as a Source Data File.

Cre-*loxP* recombination-dependent manner[18]. The original PA-Cre showed higher leakiness in the dark and ambient light conditions whereas the codon-optimized version PA-Cre 3.0 had no spontaneous Cre-*loxP* recombination at all in the mouse livers (Fig. 2d and Supplementary Fig. 5). Also, we found that the mice injected with PA-Cre 3.0 showed higher recombination efficiency and tdTomato reporter expressions than the original PA-Cre. These results reveal that PA-Cre 3.0 is applicable for blue-light-dependent Cre-*loxP* recombination in the single *loxP* pair target in mouse model in vivo. Also, this result suggests that bioluminescence quantification in live mouse livers using Luc reporter[10] might not be as reliable and appropriate as tdTomato fluorescent reporter in individual liver cells in order to detect such leak activity of Cre-*loxP* recombination in vivo tissues.

**The mechanism underlying reduced dark leak with codon optimization.** To elucidate the molecular mechanisms by which background leakiness is reduced following codon optimization, we

compared mRNA and protein expression levels of Magnets and Magnets-opti (Fig. 3a, b). First, we constructed CAG-Magnets-HA and CAG-Magnets-opti-HA plasmids, which inserted HA-tag into the carboxyl-terminus of PA-Cre, which allowed us to quantify the expression of PA-Cre proteins using HA-tag blotting (Fig. 3a). This was primarily done due to non-specific bands seen on blots from cell lysates probed with Cre antibody (Fig. 2b). First, we compared the effect of CMV and CAG promoters on the protein expressions of Magnets and Magnets-opti in HEK 293T cells. We found that there was no difference in Magnets protein between CMV and CAG promoters while CAG promoter increased Magnets-opti expression (Supplementary Fig. 6a). Next, via qPCR, we found that Magnets-opti mRNA expression was significantly lower (~40–50%) than Magnets (Fig. 3c). We then adjusted the cellular transfection in accordance with Magnets-opti transcript levels using different plasmid DNA amounts of Magnets and loaded same amount of PA-Cre proteins in Magnets (lane 3; Magnets, transfection DNA, 40% plus empty vector, 60%) and Magnets-opti (lane 4;

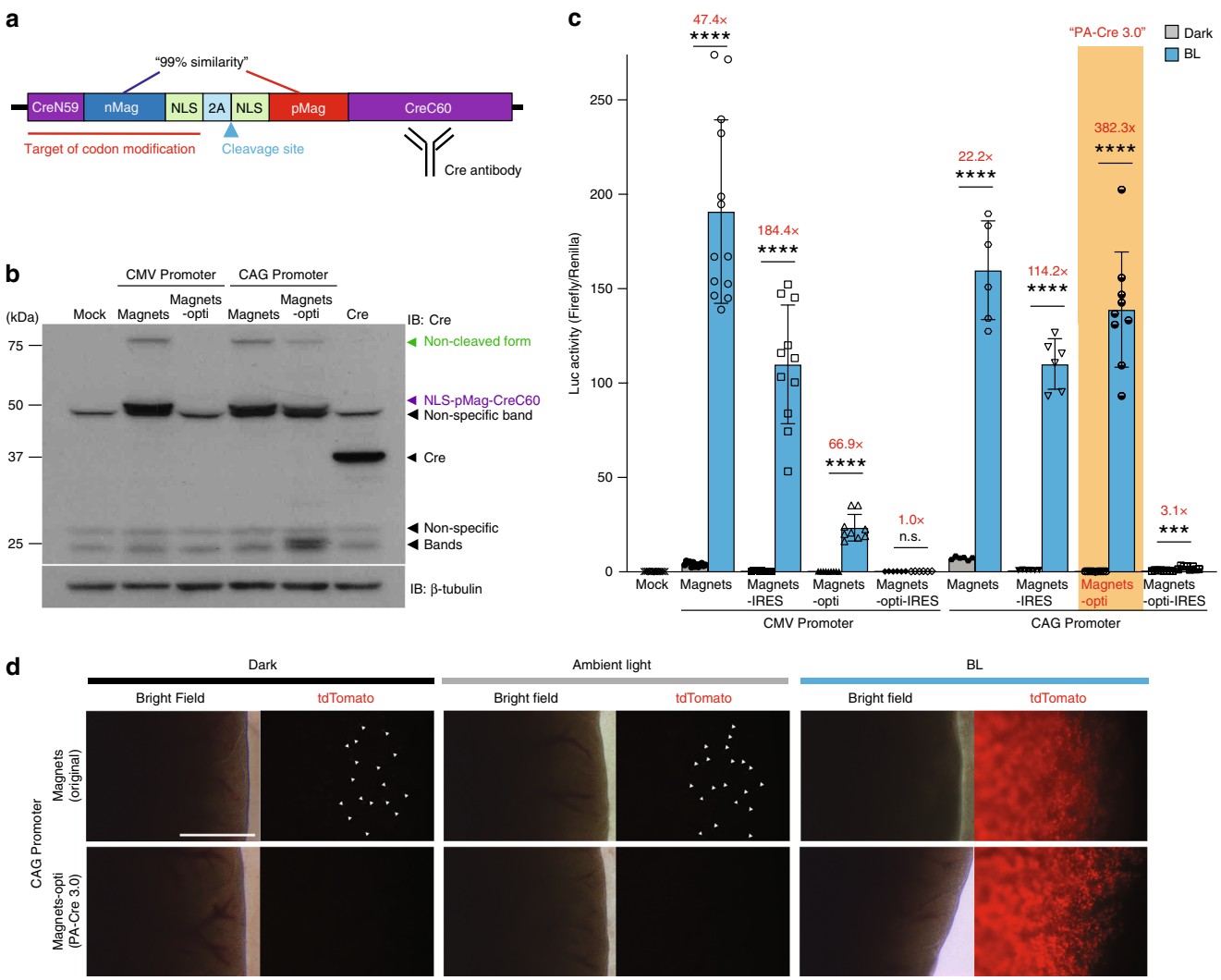

**Fig. 2 Development and characterization of PA-Cre 3.0. a** Schematic representation of PA-Cre components indicating codon modification site, 2A cleavage site and Cre antibody recognition site. **b** Representative western blotting images to compare CMV and CAG promoters for testing Magnets and Magnets codon-optimized PA-Cre (Magnets-opti). All protein expressions were accomplished in the dark condition (single independent experiments). **c** Comparisons of the original Magnets- and Magnets-opti-based PA-Cre using CMV or CAG promoter with 2A or IRES sequences (Magnets-IRES, Magnets-opti-IRES) using dual Luc assay. Luc assays were conducted in HEK 293T with double-floxed inverted Fluc. The upper diagram shows an illumination protocol used for the assays (blue LED, 447.5 nm, 8.28 W/m$^2$, repeated 20 s light and 60 s dark for 12 h) (n.s. not significant, ***$P \leqq 0.0005$; ****$P < 0.0001$; dark v.s. BL using Two-tailed $t$ test, Biologically independent samples; Mock, CMV-Magnets: $n = 13$, CMV-Magnets-IRES $n = 11$, CMV-Magnets-opti $n = 9$, CMV-Magnets-opti-IRES, CAG-Magnets, CAG-Magnets-IRES: $n = 6$, CAG-Magnets-opti $n = 9$, CAG-Magnets-opti-IRES2 $n = 8$, mean ± s.d.). Red values show fold induction of Luc with BL compared with dark. **d** Comparison of CAG-driven Magnets-opti-based PA-Cre (PA-Cre 3.0) with the original version of PA-Cre (Magnets-based) using fluorescence imaging in mouse livers in vivo. Fluorescence liver imaging of tdTomato expression was conducted in Rosa26$^{Ai14/WT}$ (Ai14: Floxed-tdTomato heterozygote) live mice transiently injected with CAG-Magnets or CAG-Magnets-opti plasmids, respectively in the dark, ambient light and under blue-light illumination (470 ± 20 nm, 200 W/m$^2$, 16 h continuous) ($n = 2$–6 mice/group, Scale bar: 1 mm). White arrowheads show tdTomato-positive cells in the livers in the dark and ambient light conditions. Source data are provided as a Source Data File.

Magnets-opti, transfection DNA, 100%) (Fig. 3d and Supplementary Fig. 6b). There are no significant differences between Magnets 40% and Magnets-opti 100% in cleaved and non-cleaved PA-Cre forms (Fig. 3e, f and Supplementary Fig. 6c–e). However, the dark leak was still significantly higher in Magnets 40 than Magnets-opti 100 (Fig. 3g and Supplementary Fig. 6f). Also, dark leak increases along with protein amount in Magnets (Supplementary Fig. 6f). These results suggest that the primary cause of dark leak in Magnets is from excess expression of mRNA and protein of PA-Cre constructs, inducing significant dark leak.

**Development of in vivo mouse applications of PA-Cre 3.0.** To apply PA-Cre 3.0 to mammalian primary cells, we transduced this

construct into mouse primary neural progenitor cells isolated from Ai14: Floxed-tdTomato heterozygous reporter mouse brains. To express PA-Cre 3.0, we applied adeno-associated virus (AAV) using the neuronal activity-dependent promoter, RAM element, and a chemical-controllable system, Tet-off[19] (Fig. 4a). Under blue-light conditions with phorbol 12-myristate 13-acetate (PMA), a PKC activator, or high KCl stimuli as depolarization induction, we observed successful Cre-loxP recombination mediated by AAV-RAM-Tet-off-PA-Cre 3.0 in Ai14 mouse-derived neurons (Supplementary Fig. 7a, b). In addition, we confirmed that doxycycline addition prevented light-dependent Cre-loxP recombination via the Tet-off system in vitro. As additional experiments, we tested a doxycycline-fed condition

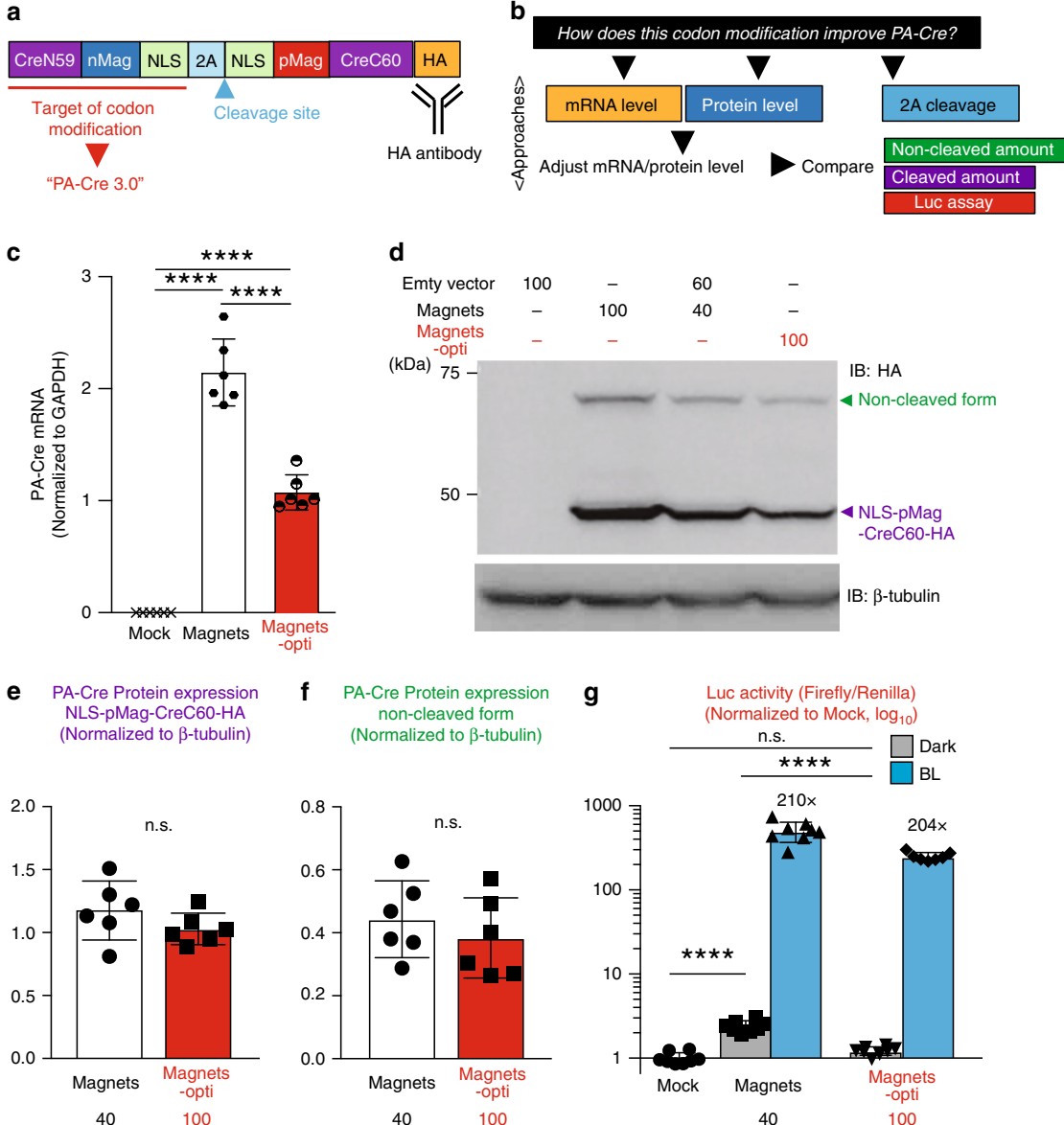

**Fig. 3 Codon modification alters mRNA and protein expression and reduces dark leak in PA-Cre. a** Schematic representation of PA-Cre construct fused with HA-tag. HA-tag was inserted into the carboxyl-terminus of PA-Cre construct. Codons of CreN59-nMag-NLS component were modified in PA-Cre 3.0 (Magnets-opti). **b** Experimental Schematic to elucidate the mechanisms underlying PA-Cre 3.0 improvement in dark leak with codon modification. First, we quantified mRNA expression level of Magnets and Magnets-opti. Second, we measured the protein expression level. In accordance with these results, we modified transfection amount to adjust PA-Cre protein expression levels in Magnets and Magnets-opti and compared the loxP recombination efficiency using luciferase assay. **c** Quantification of CAG-driven PA-Cre transcripts in HEK 293T cells transfected with same amount of Mock (empty vector), Magnets-HA and Magnets-opti-HA plasmids using quantitative RT-PCR (****$P < 0.0001$; one-way ANOVA with multiple comparison, $n = 6$ biologically independent samples, mean ± s.d.). **d** Representative western blotting images of Magnets- and Magnets-opti-transfected HEK 293T cells. CAG-driven HA-tagged PA-Cre proteins were detected by using anti-HA antibody. Loading schematics showed transfection DNA amounts of PA-Cre plasmids. Empty vector was used to adjust total amount of transfection for the cells. Protein expression of Magnets was adjusted to Magnets-opti expression, modifying Magnets plasmid DNA amount in a series of transfections (Fig. S6b). Top bands (~75 kDa), non-cleaved form of PA-Cre; bottom bands, cleaved carboxyl-terminal portion of PA-Cre construct (NLS-pMag-CreC60-HA). β-tubulin was examined as an internal control in the cells ($n = 2$ biologically independent experiments). **e** Quantification of protein expression of cleaved component of PA-Cre, NLS-pMag-CreC60-HA. (n.s. not significant, Magnets 40 v.s. Magnets-opti 100; Two-tailed $t$ test, $n = 6$ biologically independent samples, mean ± s.d.). **f** Quantification of protein expression of non-cleaved PA-Cre form. (n.s. not significant, Magnets 40 v.s. Magnets-opti 100; Two-tailed $t$ test, $n = 6$ biologically independent samples, mean ± s.d.). **g** Comparison of PA-Cre activities between Magnets and Magnets-opti using the same protein expression amounts. Luc assays were conducted with double-floxed inverted Fluc reporter in HEK 293T cells. Transfected cells were kept in the dark or under blue light (blue LED, 447.5 nm, 8.28 W/m², repeated 20 s light and 60 s dark for 12 h). The black numbers show fold-induction value (n.s. not significant, ****$P < 0.0001$; one-way ANOVA with multiple comparison among Mock, Magnets dark, and Magnets-opti dark, $n = 8$ biologically independent samples, mean ± s.d.). Source data are provided as a Source Data File.

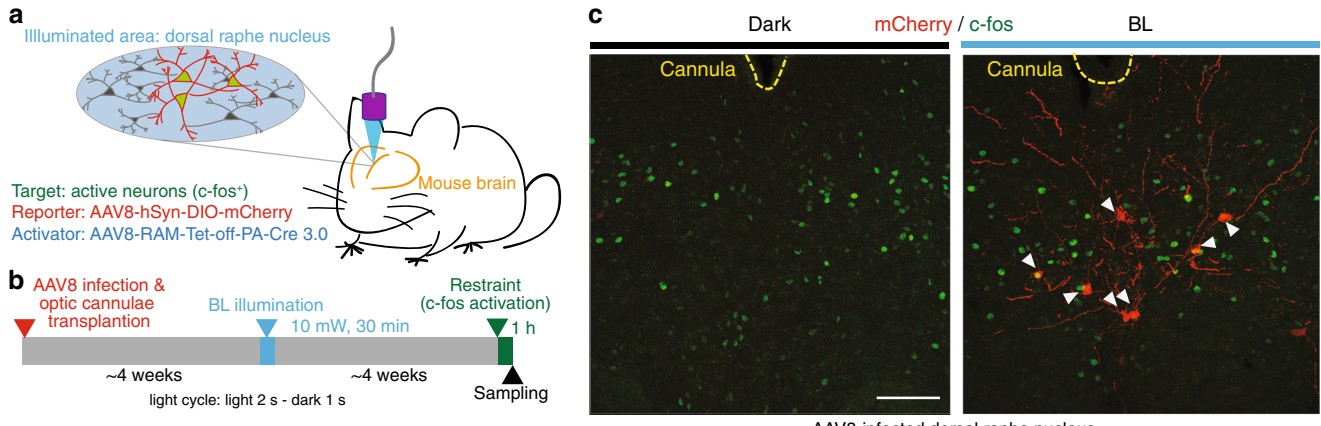

**Fig. 4 PA-Cre 3.0 AAV in vivo applications in mice. a** Schematic representation of AAV-RAM-Tet-off-PA-Cre 3.0 application for targeting active neurons in mouse dorsal raphe nucleus using RAM element and optic cannula. **b** Experimental time course of AAV-RAM-Tet-off-PA-Cre 3.0 application in mouse dorsal raphe nucleus with restraint, which can activate c-fos, and BL illumination. The mouse groups were kept in doxycycline-free condition. **c** Representative immunofluorescent images of red fluorescent protein, mCherry (red) and neuronal activity marker, c-fos (green), in mouse dorsal raphe nucleus infected with AAV-RAM-Tet-off-PA-Cre 3.0 viruses in dark ($n = 2$), and BL illumination ($n = 3$). Optic cannulas are shown in yellow dashed lines. White arrowheads show c-fos- and mCherry-positive neurons in the area. Scale bar, 100 μm.

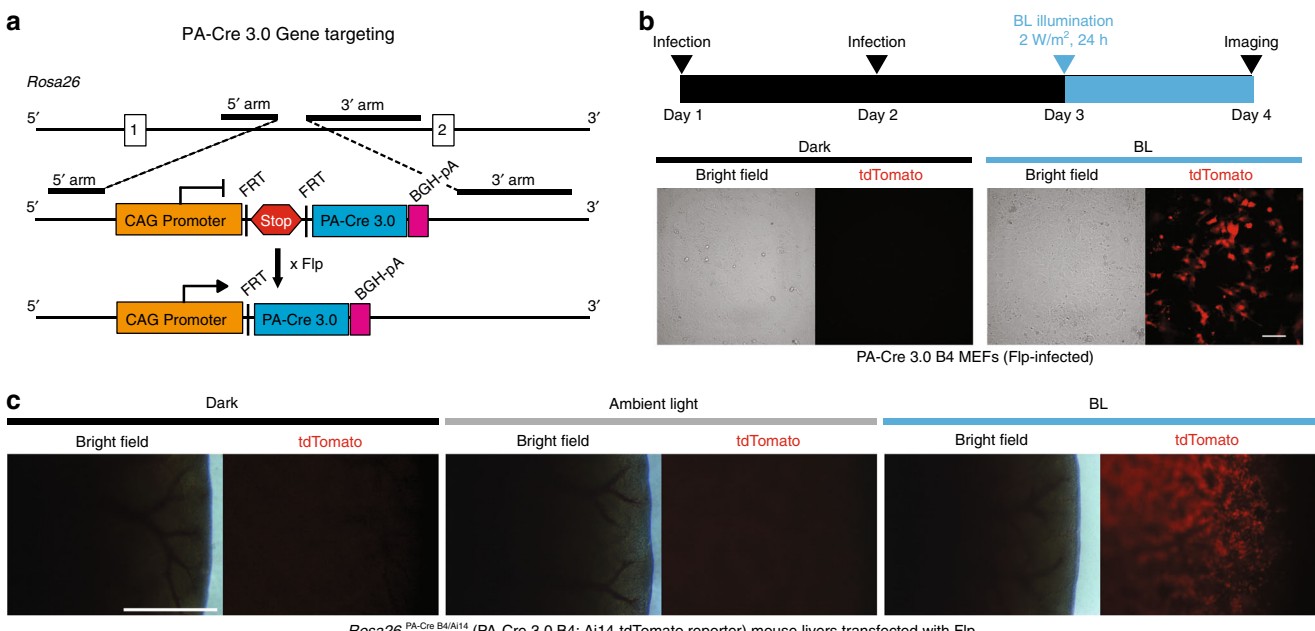

**Fig. 5 Cre recombination induced by blue light in vivo PA-Cre 3.0 mouse. a** Schematic representation of PA-Cre 3.0 mouse targeting strategy. PA-Cre 3.0 knocked-in to *Gt(ROSA)26Sor* (*Rosa26*) locus with CAG promoter and stop sequence added between FRT cassettes. Flp-mediated excision of the FRT-flanked stop cassette (Stop) induces PA-Cre 3.0 expression. **b** Representative bright field and tdTomato fluorescent images of mouse embryonic fibroblast cells derived from *Rosa26^PA-Cre B4/Ai14* mouse line illuminated with blue light. Diagram showed the experimental time course. The CAG-Flpe lentivirus was infected at the serial 2 days. The following day, the cells were illuminated for 24 h (470 ± 20 nm, 2 W/m², 24 h continuous) ($n = 3$, Scale bar: 100 μm). **c** Representative liver images freshly isolated from *Rosa26^PA-Cre B4/Ai14* mice. CAG promoter-mediated Flp plasmids were transfected into PA-Cre 3.0 B4 mouse livers using HTV method. The mice were maintained in the dark, ambient light, or under blue-light illumination (470 ± 20 nm, 200 W/m², 16 h continuous) ($n = 3$–4 mice/group, Scale bar: 1 mm).

and light stimulation of mouse brain amygdala, which responds to stress, and confirmed that the Tet-off system was functional in the AAV-based mouse model in vivo (Supplementary Fig. 7c) and is consistent with the in vitro results in the primary neural cells (Supplementary Fig. 7a, b). Furthermore, we applied the AAV-RAM-Tet-off-PA-Cre 3.0 viruses into the mouse brain dorsal raphe nucleus, which also responds stress, together with mCherry reporter viruses, and fibrotic light sources. Under blue-light conditions following AAV infection, we observed Cre-lox recombination in *c-fos*-positive neurons in the mouse brain region while there was no Cre-*loxP* recombination observed in mice kept in dark (Fig. 4b, c). These results reveal that AAV-RAM-Tet-off-PA-Cre 3.0 virus is a useful resource applicable for mouse brains in vivo.

To better facilitate in vivo studies, we generated a new mouse line by inserting PA-Cre 3.0 construct in the *Rosa26* locus (Rosa26-CAG-Frt-Stop-Frt-PA-Cre 3.0, B4 mouse embryonic stem cell clone, Fig. 5a). This mouse line enables for tissue-specific

expression of PA-Cre 3.0, following Flp-*Frt* recombination. To confirm that this targeting strategy is functional in mice, we isolated mouse embryonic fibroblasts (MEFs) from embryos of PA-Cre 3.0 B4 mouse line crossed with Ai14: Floxed-tdTomato reporter mice. We observed the expression of tdTomato red fluorescent proteins, following Flp lentiviral infection and blue-light exposure, revealing the successful Cre-*loxP* recombination without unintended leakiness in the dark condition (Fig. 5b). As an infection control, the same lentiviral system expressing yellow fluorescent proteins was tested in the MEFs (Supplementary Fig. 8). In addition, we could observe strong and specific tdTomato expressions in livers of PA-Cre 3.0 B4: Ai14 mice, following HTV injections of CAG promoter-mediated Flp plasmids into the mice (Fig. 5c and Supplementary Fig. 9).

Because mouse breeding takes time for experimental design with multiple allele uses, an all-in-one version containing both PA-Cre 3.0 and a fluorescent protein reporter will be valuable for users. Also, transient expression of PA-Cre 3.0 might be ideal as Cre recombination is transient and irreversible and continuous Cre activity might induce neuronal toxicity[20]. To accomplish this, we added *loxP* sites in the front and back of PA-Cre 3.0 construct as a self-deficient option in the *Rosa26* locus wherein the PA-Cre 3.0 can be removed by itself, following blue-light stimulation leaving only the mKate2 red fluorescent protein reporter expression (Fig. 6a). The advantage of this targeting design is an all-in-one concept for PA-Cre 3.0 and reporter expressions in the single construct. To validate this system, MEFs were isolated from the all-in-one version of PA-Cre 3.0 mouse line (clone A20) crossed with Rosa26-FLPe line. We confirmed that blue-light illumination could induce mKate2 expression, revealing that illumination induces the successful recombination by PA-Cre 3.0 and there is no spontaneous recombination in dark at all (Fig. 6b, c). The primary cortical neurons isolated from the all-in-one PA-Cre 3.0 A20 mouse line and infected with Flp lentiviruses also showed mKate2 expression with blue-light illumination (Fig. 6d, e). The results demonstrate that the all-in-one PA-Cre 3.0 A20 mouse line is a useful and reliable resource in mouse genetic study.

## Discussion

In this study, we have developed a newly improved photo-activatable Magnets-based Cre recombinase named PA-Cre 3.0. The other photoreceptors-based PA-Cre systems have shown to have disadvantages with dark leakiness as well as low recombination efficiency (Fig. 1b). Our new version, PA-Cre 3.0, has significantly addressed the issues. The results (Fig. 3 and Supplementary Fig. 6) suggest that the molecular mechanism underlying PA-Cre improvement is mainly associated with protein expression level during PA-Cre transcription and translation, but not with P2A-mediated self-cleaving efficiency while the accumulated non-cleaved form might become an additional source of dark leak (Supplementary Fig. 2). Particularly, excess expression of PA-Cre proteins may induce spontaneous nMag–pMag and/or CreN-CreC complementation, resulting in leakiness of Cre-*loxP* recombination in dark. In the original Magnets-based PA-Cre, which has higher dark leak, the highest protein expression has been accomplished using codon modification[10]. Therefore, there was no difference in CMV-driven Magnets protein expression from ones driven by a CAG promoter, which could increase Magnets-opti protein (Supplementary Fig. 6a). Interestingly, we found that Magnets-opti, PA-Cre 3.0, has a unique advantage of minimizing the dark leak because dark leak of Magnets-opti (transfection DNA: 100%) was significantly lower than Magnets (transfection DNA: 40%), while there was no significant difference in the protein amount of cleaved and non-cleaved forms between Magnets-opti and

Magnets (Fig. 3e–g). In particular, dark leak in Magnets-opti did not increase even while mRNA and protein increased (transfection: 75 and 100%) (Supplementary Fig. 6f). According to a previous report[21], codons used in Magnets might affect protein folding of PA-Cre and increase spontaneous nMag–pMag/CreN59-CreC60 associations, which induces dark leak. Our findings demonstrate that codon modification is crucial for developing and optimizing genetically encoded tools.

Considering users' convenience to optimize the amount of blue light needed for single genomic Cre-*lox* recombination, we infected PA-Cre 3.0 to MEFs isolated from Ai14: Floxed-tdTomato heterozygous reporter mice. In the previous study, reporter plasmids were used. However, a single copy target of *loxP* pair would be more appropriate to validate the PA-Cre system. We found that at least 5 min of blue-light stimulation was required for Cre-*loxP* recombination (Supplementary Fig. 10). In addition, we also repeated the 3 h blue-light stimulation under varying amounts of light intensity and we found 1.5 W and greater induced recombination in MEFs. Also, the result of PA-Cre 3.0 application using AAV8 in mouse brains in vivo (Fig. 4 and Supplementary Fig. 7) suggests that standard in vivo opto-genetic illumination protocol[22] is applicable for activating PA-Cre 3.0.

For in vivo applications, PA-Cre 3.0 B4 and A20 mouse lines were generated and their germ-line transmission was confirmed. We successfully demonstrated the effectiveness of PA-Cre 3.0 in the isolated primary neural cells and MEFs as well as the livers and AAV-infected mouse brains in vivo (Figs. 4–6). As a valuable resource in the research community, the PA-Cre 3.0 mouse lines and viruses will be useful and shared to facilitate spatiotemporal Cre-*loxP* recombination for further biological applications.

## Methods

**Plasmid construction**. Plasmid DNA constructs were generated using standard methods with restriction enzymes (New England BioLabs), DNA ligase (MightMix, TaKaRa) and polymerase chain reaction (PCR) with polymerase (Thermo-Fisher). The transient expression vector pcDNA3.1 (Life Technologies) was used with the CMV promoter, and lentiviral vector LV-SD vector (Addgene, #12105, LV-Cre-SD, no longer available currently) was used as CAG promoter-mediated vector. The CMV promoter-mediated Magnets-based PA-Cre was developed in our previous study[10]. We synthesized the codon-modified CreN59, nMag and NLS sequences (Integrated DNA Technologies, IDT) and inserted into *Nhe* I/*BamH* I sites instead of the original codon sequences without any amino acid substitution. The CRY2/CIB1 vector was kindly gifted from Dr. C. Tucker (Univ. Colorado). To construct the iLID/SspB-based PA-Cre plasmid, the full-length iLID/SspB-based PA-Cre construct was synthesized and inserted *Nhe* I sites of pcDNA3.1 vector. To construct the FKF1/GI vectors, we used pcDNA3-NLS-Gal4DBD-NLOV(H105L)-HA vector and pcDNA3-FLAG-GI-VP16 (ref. [13]) as backbone vectors. We cut out the Gal4DBD and inserted NLS-CreN59 into *Kpn* I/*Not* I sites of pcDNA3-NLS-Gal4DBD-NLOV(H105L)-HA. We cut out VP16 and inserted CreC60 into *Not* I/*Xba* I sites of pcDNA3-FLAG-GI-VP16 vector. To construct the CreERT2 plasmid, we used pCAG-CreERT2 (Addgene, #14797) as a PCR template and inserted CreERT2 sequence into *EcoR* I/*Not* I sites of pcDNA 3.1 vector. To construct the P2A mutant and F2A, we used CMV-mediated Magnets-based PA-Cre[10] as a backbone vector and inserted the synthesized constructs into *BamH* I/*Xho* I sites. To construct the CMV-mediated Magnets- Magnets-opti-based PA-Cre IRES versions, we used CRY2/CIB1-based version, called PA-Cre 2.0 (ref. [11]), as a PCR template for IRES sequences and inserted IRES sequence into *BamH* I/*Xho* I sites of each vector instead of P2A sequence. To construct the CAG-mediated Magnets, Magnets-IRES, Magnets-opti, and Magnets-opti-IRES vectors, we cut out Magnets, Magnets-IRES, Magnets-opti and Magnets-opti-IRES portions from CMV-Magnets, CMV-Magnets-IRES, CMV-Magnets-opti, and CMV-Magnets-opti-IRES, respectively. And then we inserted each into *Nhe* I/*Xba* I sites of LV-SD vector that contains homemade multi cloning sites (MCS), *EcoR*I-*Asc*I-*Nhe*I-*Pac*I-*Xba*I-*Xho*I, instead of *EcoR*I-NLS-Cre-*Xho*I. To construct the AAV-PA-Cre 3.0 vector, we used pRAM-d2TTA-TRE-MCS-WPRE-pA (Addgene #63931) as a backbone vector. We inserted Magnets-opti version (PA-Cre 3.0) sequence into *Nhe* I/*Asc* I sites and deleted the WPRE sequences in order to maintain AAV packaging capacity. To construct the mouse targeting vector, we used STOP-eGFP-ROSA26TV (Addgene, #11739) as a backbone vector. We subcloned CAG promoter_FRT_TKneo-polyA_FRT_*loxP*_Magnets-opti_WPRE-polyA_*loxP*_Kozak-NLS-mKate2-STOP sequence and used it as an all-in-one targeting vector (named, #A20, a positive mouse embryonic stem cell clone). We

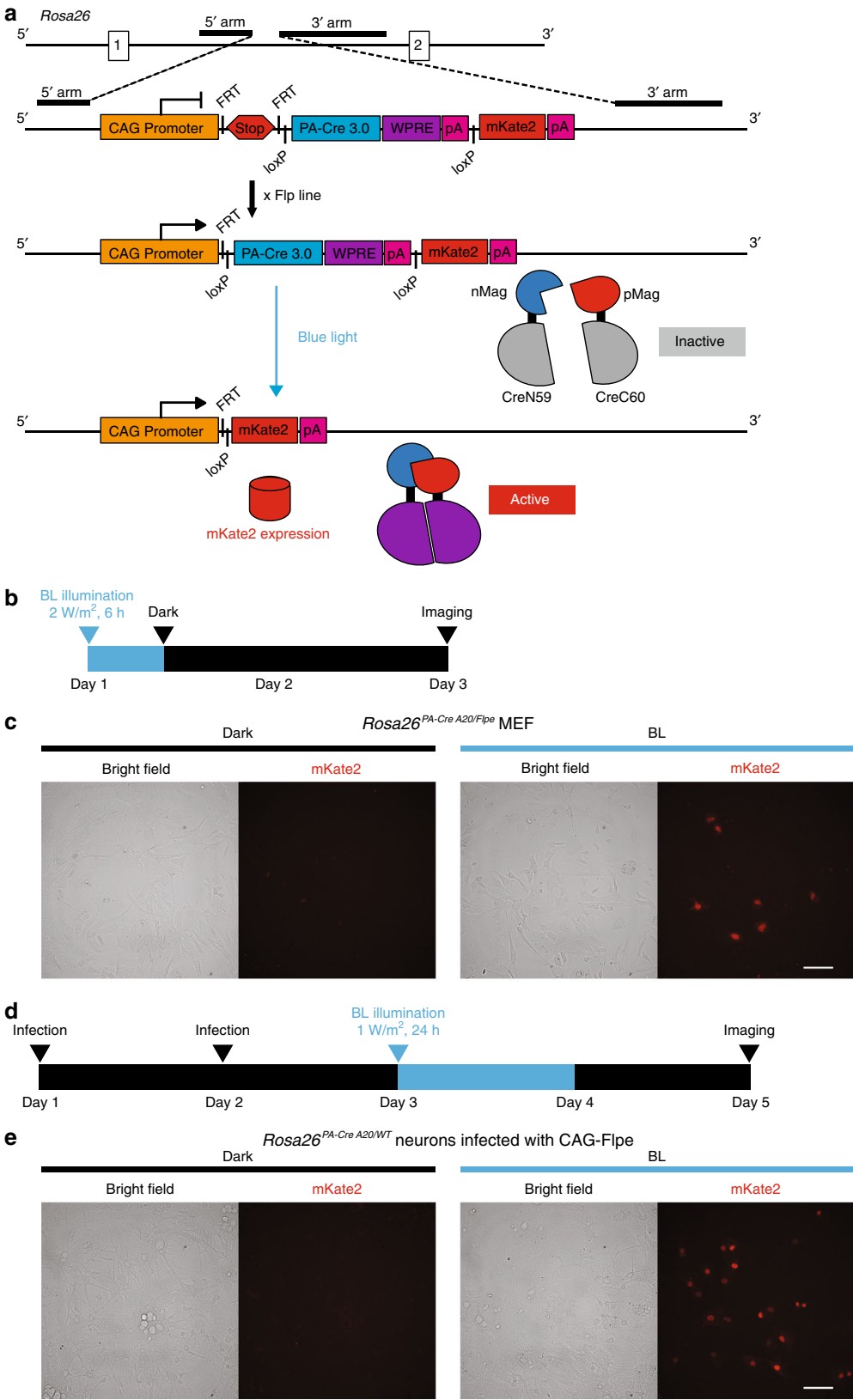

also subcloned CAG promoter_FRT_
TKneo-polyA_FRT_Magnets-opti sequence and used it as a targeting vector
(named B4). To construct CMV-Magnets-HA, CMV-Magnets-opti-HA, CMV-
Magnets-opti-IRES-HA, CAG-Magnets-HA and CAG-Magnets-opti-HA, we syn-
thesized HA-tag; HA, 5′-GCTGGCTCCTACCCATACGATGTTCCAGATTACG
CT-3′ and inserted to the carboxyl terminal of each vector, respectively.

**Animals**. Experiments using mice were performed in accordance with the
Guidelines for Care and Use of Laboratory Animals as stated by Columbia Uni-
versity IACUC. The *Rosa26^Ai14/Ai14* mouse (B6.Cg-Gt(ROSA)26Sor^tm14(CAG-tdTo-
mato)Hze/J; The Jackson Laboratory, Stock no: 007914) were purchased from the
Jackson Laboratory and crossed with C57BL/6 mice to generate *Rosa26^Ai14/WT*.
*Rosa26^Ai14/WT* mice were sacrificed for HTV injection at 5–6-week-old and also

**Fig. 6 Generation and validation of all-in-one version of PA-Cre 3.0 mouse line. a** Schematic representation of PA-Cre 3.0 "All-in-One" mice targeting strategy. CAG promoter and PA-Cre 3.0 knocked-in to *Gt(ROSA)26Sor (Rosa26)* locus was conducted together with stop cassette inserted between FRT cassettes and nuclear-localizing signal (NLS)-fused mKate2 red fluorescent protein. The PA-Cre 3.0 is located between *loxP* pair. Flp-mediated excision of the FRT-flanked stop cassette results in the expression of PA-Cre 3.0. After the blue-light stimulation and PA-Cre activation, the PA-Cre is self-cleaved. PA-Cre-mediated excision of the PA-Cre 3.0 cassette including WPRE element induces NLS-mKate2 expression. The mKate2 red fluorescence signals located in nuclei can be used as a Cre-*loxP* recombination reporter. **b** Schematic representation of experimental time course for MEFs derived from *Rosa26^PACre A20/FLPe* mice. The cells were illuminated for 6 h (470 ± 20 nm, 2 W/m$^2$, 6 h continuous). The cells were imaged 42 h after illumination. **c** Representative mKate2 fluorescence images of PA-Cre 3.0 A20 MEFs (Left images, the bright field and right images, the mKate2 red fluorescence; $n = 3$, Scale bar: 100 μm). **d** Schematic representation of experimental time course for cortical neurons derived from *Rosa26^PACre A20/WT* mouse brains with illumination. The CAG-Flpe lentivirus was infected at serial 2 days. At the following day (day 3), the cells were illuminated for 24 h (470 ± 20 nm, 1 W/m$^2$, 24 h continuous). **e** Representative mKate2 fluorescence images of PA-Cre 3.0 A20 neurons (Left images, the bright field and right images, the mKate2 red fluorescence; $n = 3$, Scale bar: 100 μm).

used for MEFs isolation at E11.5–13.5 and used for neuronal progenitor cells (NPCs) isolation at E12.5. To generate PA-Cre chimeric mice, we established mouse embryonic stem cells (mESCs) lines (A20 and B4) with integration of Magnets-opti into *Rosa26* locus. mESC injection into blastocysts for chimeric mouse generation was accomplished at the transgenic animal facility in Columbia University Medical Center. The PA-Cre chimeric mice were crossed with C57BL/6 mice and confirmed the germ-line transmission by genotyping PCR. F1, F2, and F3 generations named *Rosa26^PA-Cre A20 or PA-Cre B4/WT* maintained by crossing with C57BL/6 mice. The *Rosa26^PA-Cre B4/WT* mice were crossed with *Rosa26^Ai14/Ai14* mice to generate *Rosa26^PA-CreB4/Ai14*. *Rosa26^PA-CreB4/Ai14* mice used for HTV injection at 5–6-week-old and also sacrificed to isolate the MEF at E11.5–13.5. The *Rosa26^PA-Cre A20/WT* mice were used cortical neuron isolation at P1-4. The *Rosa26^PA-Cre A20/WT* mice were crossed with *Rosa26^FLPe/FLPe* mice (B6.129S4-*Gt (ROSA)26Sor^tm1(FLP1)Dym*/RainJ, JAX# 009086) to generate *Rosa26^PA-Cre A20/FLPe*. *Rosa26^PA-Cre A20/FLPe* mice used for MEF isolation at E11.5–14.5.

**Cell culture.** Human embryonic kidney (HEK) 293T cells (ATCC, #CRL-3216) were cultured in Dulbecco's Modified Eagle Media (DMEM, Thermo-Fisher) supplemented with 1% GlutaMax I (Thermo-Fisher) and 100 U/ml penicillin and 100 μg/ml streptomycin (P.S., Thermo-Fisher) and 10% fetal bovine serum (FBS, not heat-inactivated, HyClone, Thermo-Fisher). HEK 293T cells were passaged using 0.05% trypsin + 0.03% EDTA solution (Thermo-Fisher). mESCs derived from C57BL/6 and 129SV/EV hybrid mouse (by Columbia Transgenic mouse facility, Chyuan-Sheng Lin) were cultured on Mitomycin C-treated DR4 feeder cells. The cells were maintained DMEM (Thermo-Fisher) supplemented with 20% FBS, nonessential amino acids (NEAA, Thermo-Fisher), 1% GlutaMax I, 1% nucleosides (Millipore-Sigma, ES-008-D), and 2-mercaptoethanol (Sigma-Aldrich). The targeting vector was electroporated into undifferentiated mESCs. Twenty-four hours after electroporation, the medium was replaced by selection medium containing the 50 μg/mL G418 (Clontech) for 7 days to establish the targeting mESC lines. The targeting plasmid integration was confirmed by PCR with following primers: 5′ GGACTAGGGCTGCGTGAGTCTCTGA (forward) and GGCGTTAC TATGGGAACATACGTC (reverse). The targeting mESCs were used for chimeric mice generation. MEFs cells were isolated from E11.5 to E13.5 mouse embryos by using 0.25% Trypsin-EDTA (Thermo-Fisher) and DNase solution (Worthington). MEF cells were cultured in DMEM supplemented with 10% FBS, 1% GlutaMax I, and P.S. MEF cells were passaged using 0.25% trypsin + 0.03% EDTA. NPCs were isolated from brains of E12.5 mouse embryos and dissociated with accutase (Thermo-Fisher) and DNase I solution (Stem Cell Technologies). NPCs were proliferated with neural induction medium, its supplement, and Advanced DMEM/F12 medium (both from Thermo-Fisher) and then differentiated to neurons by Neurobasal Plus medium (Thermo-Fisher) supplemented with B27 (Thermo-Fisher), BDNF (PeproTech), NT3 (PeproTech), 1% GlutaMax I, and P.S. Cortical neurons were isolated from P1 to P4 mouse by using Papain, Hibernate A, and DNase solution. The cells were harvested on PDL/laminin-coated dish with Neurobasal medium, B27, and 1% FBS. The following day, the medium was changed to Neurobasal medium supplemented with B27, 1% GlutaMax I, HEPES (all, Thermo-Fisher), and P.S. All cells were cultured under normoxia (20% O$_2$, 5% CO$_2$, at 37 °C) using HERAcell CO$_2$/O$_2$ incubator (Thermo-Fisher).

**Luciferase assay.** HEK 293T cells were plated at $0.5 × 10^5$ cells/well in 24 well plates (Corning) coated with poly-ornithine (Sigma-Aldrich). The following day, the cells were transfected with cDNAs encoding PA-Cre along with *Firefly* Luc reporter and *Renilla* Luc using X-treameGENE9 reagent (Sigma-Aldrich). The standard transfection ratio of PA-Cre:double-flox-inverted-FLuc:HSV-TK-Renilla Luc was as follows: 1:9:0.1. The cells were then exposed to blue light (total 12 h, 20 s light/60 s dark cycle, homemade device[13]) 36 h after transfection at 37 °C in a CO$_2$ incubator (Thermo-Fisher). In case of tamoxifen-treated Mock or CreERT2, the cells were treated with 1 μM tamoxifen (Sigma-Aldrich) 36 h after transfection. Luc activity levels were assayed right after illumination was terminated using Dual Luciferase assay kit and Veritas 96-well luminometer (Promega) following the manufacturer's instructions.

**Fluorescence imaging.** HEK 293T cells were plated at $2 × 10^5$ cells/dish in 3.5 cm glass bottom dish (MatTek) coated with poly-ornithine. The following day, the cells were transfected with cDNAs encoding PA-Cre along with Floxed-mCherry reporter using X-treameGENE9 reagent. The standard transfection ratio is PA-Cre and Floxed-mCherry at 1:9 ratio. The cells were then exposed to blue light (total 12 h, 20 s light/60 s dark cycle, homemade device) 36 h after transfection at 37 °C in a CO$_2$ incubator. The cells were imaged with TxRed filters and Nikon NIS using the custom Nikon microscope Ti-E immediately after illumination was completed. The Cre-*loxP* recombination efficiency was analyzed by ImageJ software (NIH) upon the red fluorescent signals. MEF cells, NPCs, and cortical neurons were also imaged with TxRed filters and Nikon NIS using the custom Nikon microscope Ti-E.

**Western blotting.** The protein samples were extracted from HEK 293T cells transfected with each plasmid (pcDNA3.1, Cre, CMV-Magnets, CMV-Magnets-F2A, CMV-Magnets-P2A mutant, CMV-Magnets-opti, CAG-Magnets, CAG-Magnets-opti, CMV-Magnets-opti-HA, CMV-Magnets-opti-HA, CMV-Magnets-opti-IRES-HA, CMV-Magnets-HA, and CAG-Magnets-opti-HA) using extraction buffer (50 mM Tris-HCl (pH7.4), 150 mM NaCl, 1% TritonX-100, and 1% protease inhibitor cocktails). Cells were kept in dark and harvested 48 h after transfection. The samples were placed on ice for 30 min to complete permeabilization. Lysate were centrifuged at 14,000 rpm for 10 min at 4 °C to remove cell debris. The supernatant samples were moved to new tubes. An equal volume of 2× sodium dodecyl sulfate (SDS) sample buffer (8M Urea, 40 mM Tris-HCl pH7.4, 2% SDS, 10% 2-mercaptoethanol, and 0.01% bromophenol blue) was added to the lysate samples and boiled for 5 min on heat block (~95 °C). The protein samples were subjected to SDS-polyacrylamide gel electrophoresis using Tris-Glycine-based gels (Bio-Rad) containing 10% Acrylamide-Bis (Fisher Scientific), and then electro-transferred Polyvinylidene difluoride (PVDF) membranes using XCell SureLock® Mini-Cell and XCell™ Blot Module system (Life Technologies). The membranes were blocked with 5% or 10% nonfat milk powder in TBS-T (Tris-based saline solution with Tween, 50 mM Tris-HCl, 150 mM NaCl, and 0.1% Tween) or 1% bovine serum albumin (BSA) in TBS for overnight at 4 °C or 1–4 h at room temperature (RT). Then, the membranes were incubated with following primary antibodies: anti-Cre (Abcam, #ab24608, 1:1000 dilution in TBS-T solution with 5% BSA), anti-HA (Roche, #11867423001, 1:1000 in TBS solution with 1% BSA), and anti-β-tubulin (Sigma-Aldrich, #T5201, 1:8000 dilution in TBS-T solution with 5% BSA or in TBS solution with 1% BSA) for 1 h at RT followed by an incubation of the corresponding secondary antibody (Thermo-Fisher, Pierce, anti-rabbit, #31460; anti-mouse #31430; anti-rat, #31470 1:8000 dilution in TBS-T with 5% nonfat skim milk or in TBS solution with 1% BSA) for 1 h at RT. The protein signals were detected using Pierce ECL western blotting substrate (Thermo-Fisher) by exposing to X-ray films (Thermo-Fisher) in a dark room. The original blots are also available in scanned film images provided in the Supplementary Source Data file.

**Quantitative RT-PCR.** The RNA samples were extracted from HEK 293T cells transfected with each plasmid (pcDNA3.1, CMV-Magnets-HA, CMV-Magnets-opti-HA, CMV-Magnets-opti-IRES-HA, CAG-Magnets, CAG-Magnets-HA, and CAG-Magnets-opti-HA) using RLT buffer (Qiagen). Cells were kept in dark and harvested 48 h after transfection. The RNA was prepared using RNeasy Mini kit and RNase-Free DNase set (Qiagen). cDNA was synthesized using the SuperScript III First-Strand Synthesis System for RT-PCR (Thermo-Fisher). The cDNA (21 μl) was diluted with DNase-free water (Thermo-Fisher) at a ratio of 1:4 and 1 μl of the samples was used for qPCR analysis. SYBR Advantage qPCR Premix (Clontech/ TaKaRa Bio) and StepOnePlus real time PCR systems (Thermo-Fisher) were used for qPCR. The primer sets for detecting the CreC60 and GAPDH transcripts were as follows: CreC60, Forward 5′-GAGATACCTGGCCTGGTCTGG-3′, Reverse 5′-ACATTGGTCCAGCCACCAGC-3′; GAPDH, Forward 5′-GATGACATCAAG AAGGTGGTGA-3′, Reverse 5′-GTCTACATGGCAACTGTGAGGA-3′ (oligo synthesis, IDT). The CT value of each sample at 50% of the amplification curve was used and GAPDH was used to normalize the expression of CreC60.

**Hydrodynamic tail vein injection.** We used 5–6-week-old *Rosa26^Ai14/WT* mouse or *Rosa26^PA-Cre B4/Ai14* mouse. The abdominal surface furs of mice were removed using hand trimer. The fur-removed mice were rested for at least 24 h in a cage, and

intraperitoneally injected with cDNAs encoding CAG-Magnets, CAG-Magnets-opti, Cre, or pCAG-Flpe (Addgene, #13787) using TransIT®-EE Delivery Solution (Mirus Bio LLC), respectively, according to the manufacturer's protocol. The amount of injected DNA was 10 μg per mouse. The volume of delivery solution was 0.1 mL per mouse weight (g). After the hydrodynamic injection of DNA, the mice were kept in the dark for 8 or 32 h and then illuminated with a LED source (470 ± 20 nm; 200 W/m², CCS) for 16 h in a cage. After illumination, the mice were kept in the dark to recovery for 24 h. As with ambient light illumination, the mice were kept under a standard white fluorescent light (0.05 W/m²) for 48 or 72 h. The mice were sacrificed using standard procedure approved in our animal protocol and their livers were immediately harvested for fluorescent imaging with fluorescent stereoscope (Zeiss).

**In vitro lentiviral infection**. To produce the lentivirus, the LV-SD-Magnets, LV-SD-Magnets-opti, LV-SD-Flpe or LV-SD-YFP vector were transfected together with pCMV-dr8.2 dvpr (Addgene #8455) and pCMV-VSV-G (Addgene #8454) into HEK 293T cells. Forty-eight and seventy-two hours after transfection, the supernatants of culture media were collected and concentrated by using Lenti-X concentrator (TaKaRa), following the manufacture's instruction. The lentiviral infection to MEF and cortical neuron cells were conducted once or twice as shown in the experimental procedures of figures. Blue-light illumination was started 24 h after last infection. The following day or 2 days after the illumination, reporter expression was imaged and analyzed using the custom Nikon microscope Ti-E.

**Adeno-associated viral infection**. Adeno-associated viral (AAV) production was accomplished at the virology core facility in Columbia University. The $9.5 \times 10^{10}$ vg adenovirus infected into neurons derived from NPCs in 3.5 cm film-bottom dish (Matsunami). One week after the infection, 1 μM PMA (12-myristate-13-acetate, Sigma-Aldrich) was treated with 24 h blue-light illumination. Following the PMA treatment, 67 mM KCl (Sigma-Aldrich) was treated with 6 h blue-light illumination (470 ± 20 nm, 1 W/m², continuous, CCS). The fluorescence images were obtained at the following day after the blue-light illumination using the above custom Nikon microscope. Male C57BL/6 J mice (age: 8–10 weeks old) were injected with 200 nl 50:50 mixture of AAV8-hSyn-DIO-mCherry (titer = $1.3 \times 10^{13}$, Addgene) and AAV8-RAM-PA-Cre (titer = $4.6 \times 10^{12}$ vg/ml, Columbia Vector core) into the target regions of mouse brains. Fabricated fiber optics were implanted 200 μm above injection sites[22]. Following surgery, the AAV8-infected mice were either placed on doxycycline diet (200 mg/kg, Bioserv) or regular diet and given 3 weeks to allow expression of virally delivered proteins. Blue light (470 nm, 5–7 mW) was delivered through a fiber optic at a pulse rate of "2 s on/1 s off". Three weeks later, the mice were restrained using the standard manual restraint method[23]. Specifically, the mice were placed headfirst in a ventilated 50 ml conical tube and confined with a plunger to restrict movement for 1 h. The restraint procedure was used to induce neuronal activation in the target brain area, resulting in increased c-fos⁺ neurons. The whole body of mice were perfused and fixed wit 4% paraformaldehyde for 1 h immediately after end of restraint, and then the brains were harvested for cryosection. Amygdala sections were stained with primary antibodies for c-fos (polyclonal anti-rabbit, Synaptic Systems, Cat#226003) and mCherry (monoclonal anti-rat, 1:500, Invitrogen Cat#16D7) and secondary antibodies (goat anti-rabbit Alexa 488 Cat#A11008, goat anti-rat Alexa 568, Invitrogen Cat#A11077) as well as nuclear stain Draq5 (5 μM, Thermo-Fisher Cat#62254). Dorsal raphe sections were processed as described previously[22]. Immunofluorescent images were collected on Leica DMi8 confocal microscope.

**Statistical analysis**. The statistical significance (P value) was used to compare samples with/without light exposure was determined using a F-test and a two-tailed Student's t test (Prism 7–8, GraphPad, or Excel, Microsoft). The statistical significance of the comparison of multiple samples was computed using Ordinary one-way ANOVA with multiple comparisons (Prism 7–8). The exact p value is provided in the Supplementary Source Data file.

**Reporting summary**. Further information on research design is available in the Nature Research Reporting Summary linked to this article.

## Data availability
The authors confirm that all relevant data are included in the paper or the Supplementary Information files. The source data and scanned film images for figures are provided as Source Data files. Additional information is available from the authors upon request.

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

## Acknowledgements
We thank Chandra Tucker (University of Colorado, Aurora) for sharing her PA-Cre 2.0 construct. This study is supported by CREST (Core Research for Evolutionary Science and Technology) Program (to M.S. and M.Y.) and Columbia Stem Cell Initiative (to M.Y.).

## Author contributions
M.Y. conceived of and designed this project. K.M., K.F., C.d.S-T., A.L.G-G., R.B., A.D.K., N.G., H.E.Y., S-H.E.P., G.S.C., M.C.S., and M.Y. designed and performed the experiments and analyzed the data. K.M., C-S.L. and M.Y. conducted mouse embryonic stem cell targeting and the mouse generation. K.M., R.B., and M.Y. interpret data and wrote the paper. F.K., M.S., K.L.T., E.A., and M.C.S. supervised this interdisciplinary team and proofread the paper. All co-authors approved the paper.

## Competing interests
The authors declare no competing interests.
