## [Peer Review File · Nature Communications]

Reviewers' comments:

Reviewer #1 (Remarks to the Author):

Morikawa et al., titled Photoactivatable Cre recombinase 3.0 for *in vivo* mouse applications, showed an improved version of Cre recombinase activated by blue light *in vitro* and *in vivo* systems which are insisted that almost zero background spontaneous activation unless blue light stimulation was given. Because Cre and FLP based engineered tools have a critical defect when they get a spontaneous activation without the stimulation (i.e. light, chemical), it is necessary to develop leak-free Cre recombinase system available *in vivo* systems. Especially in neuroscience research, because of the variety of cell types in the brain, cell-type-specific activation or expression of gene of interest available tools has an outstanding position in neuro tools.

The major concerns are :

1. In this manuscript, the authors show an improved version of PA-Cre, named PA-Cre 3.0, suggesting the original PA-Cre has leaky effect due to non-cleaved form from P2A sequence. Also, they optimized each sequence of Magnet pair for preventing the transgenic line from unexpected recombination in their genomes. However, **it is not clear** why the sequence optimization makes PA-Cre 3.0 free from non-cleaved form. The putative model, traffic jam, author suggested, would not be the reason. If this hypothesis is right, the author should show explainable data.

2. In figure 1f, **it is not clear** to see the anti-Cre band in lane 3 (CMV promoter-driven Magnets-opti). Supposing not expressed, no non-cleaved form should be detected and also no basal effect in the dark state. It is also linked with figure 1g in that how CMV promoter-driven Magnets-opti with blue light illumination induced luciferase activity. The author, I think, should reconsider this issue.

3. However, suspiciously, CAG promoter-driven Magnet-opti showed non-cleaved form band around 75kDa. The authors mentioned non-cleaved form is a possible reason why original PA-Cre has basal effect. Also, they suggested putative hypothesis the influence of non-cleaved form for basal effect. I mean, because as not like as other optogenetic tools, basal effect of Cre and FLP recombinase activity is critical. However, a non-cleaved form still exists in CAG promoter-driven PA-Cre. Even CAG promoter has somehow strong driving force compared to CMV, transient transfection in HEK293T cell does not fulfill the CMV vs CAG. So, the author should correct this issue with major concern #2.

4. In supplementary figure 5., figure 2e expression marker would be needed for image analysis (i.e. EGFP for PA-Cre and FLP-infected) to count PA-Cre expressed cells. In example, it is not convinced Magnets-opti-IRES vector was expressed or not.

5. **The authors should explain in figure 2a-c more detail and precisely.** For example, it is not matched in that if the author had not used Tet-off system (**No Dox+ in the illustration and figure legends at all!**), the mCherry positive cells would have existed almost all cells in the target regions. So, it is not clear, and I cannot give comments in figure 2a-c.

6. In supplementary figure 5, the authors insist higher promoter activity minimizes the leak of PA-Cre in the dark state. However, in supplementary figure 11, the authors mention excess expression possibly results spontaneous complementation of PA-Cre with mentioning in the discussion part “Particularly, excess expression weaker promoter may induce lower recombination issue.” (line 196-201). The authors should clear this issue.

6-1 The author also verified the expression level of “CMV vs CAG” when it comes to their tool. There might be no significant level, or CAG would be little more powerful than CMV from cell types to cell types (Qin *et al.*, *PLOS ONE* 2010).

Minor concerns are :

1. In figure 1a, correct 'codom' to 'codon' under the construct schematic illustration.
2. If the non-cleaved form is the major concern for basal effect of original PA-Cre, although I'm concerning the benefit of one vector system, why don't the authors test two viral vectors with viral titer optimization rather than 2A ribosomal skipping system for fortifying their hypothesis? I think if the leak problem results in excess expression, adjusting PA-Cre expression would be one of the possible optimization.
3. Correct the figure name for Supplementary Figure 2 and 4.
4. In Figure 1h and Supplementary Figure 5-6, I wonder it is OK for the liver condition of mice after the 16hr light stimulation. 20mW/mm^2 is extreme light intensity for mice causing heating effect.
5. **Describe 'Restraint' methods.**
6. The authors should explain the process of 'de-optimization' with detail. Does it mean sequence optimization for PA-Cre3.0 caused protein expression level decreased?
7. The lentiviral expression would also need expression marker like EGFP.

Reviewer #2 (Remarks to the Author):

The manuscript by Morikawa et al. describes the development and in vivo applications of the new version of photoactivatable Cre recombinase (named PA-Cre 3.0). As its previous version PA-Cre, PA-Cre 3.0 is based on the blue light-dependent dimerization system, Magnets. PA-Cre 3.0 has very low recombination activity in dark but high activity when illuminated with blue light which makes PA-Cre 3.0 quite useful in a number of genome engineering applications.

There are two major differences between the PA-Cre and the PA-Cre 3.0 systems: (1) in PA-Cre 3.0, the codons of the nMag gene are 'de-optimized', and (2) PA-Cre 3.0 performs best when relatively weaker promoters (compared to the full blown CMV) are used to express the system (the CAG promoter and the neuron specific RAM promoter were tested). Taken together, these differences, the authors suggest, reduce the 'traffic jam' of ribosomes on the PA-Cre 3.0 mRNA which leads to the more efficient production of the components of the PA-Cre 3.0 system.

The authors successfully tested PA-Cre 3.0 in two in vivo applications: in mouse liver and in neurons in the newly generated mouse line expressing PA-Cre 3.0.

Although the PA-Cre 3.0 system appears to be very useful, it seems far from being fully characterized since its activity depends on the cell type and the reporter system used to measure its activity.

The manuscript is clearly written and easy to follow. The conclusions are supported by the data. The developed PA-Cre 3.0 system will be quite useful in biomedical research.

The following are relatively minor deficiencies that need to be addressed:

The Abstract is too short (that is, not fully informative) for my taste since it does not clarify (even in general terms) what the authors mean by saying "the new version ... performs better than the previously reported ones". It would be nice to have a sentence added so the readers can understand just from the Abstract (without reading the Introduction) why the PA-Cre 3.0 system is considered better.

Fig. 1A. There seems to be a mistake in the representation of the double floxed reporter: the order of loxPs and lox2722s is incorrect (in the relative location shown there will be no deletion of the corresponding lox sites).

It is not completely clear why the authors did not experiment with the IRES-Magnets-opti constructs and the CAG promoter. On one hand, this construct did not work with the CMV promoter but on the other hand, since the nature/strength of the promoter does seem to make a difference for the functional performance of the system, it is not that apparent that the IRES-Magnets-opti construct paired with the CAG promoter will generate the same result.

Reviewer #1 (Remarks to the Author):

Morikawa et al., titled *Photoactivatable Cre recombinase 3.0 for in vivo mouse applications*, showed an improved version of Cre recombinase activated by blue light in vitro and in vivo systems which insisted that almost zero background spontaneous activation unless blue light stimulation was given. Because Cre and Flp based engineered tools have a critical defect when they get a spontaneous activation without the stimulation (i.e. light, chemical), it is necessary to develop leak-free Cre recombinase system available in vivo systems. Especially in neuroscience research, because of the variety of cell types in the brain, cell-type-specific activation or expression of gene of interest available tools has an outstanding position in neuro tools.

The major concerns are:

1. In this manuscript, the authors show an improved version of PA-Cre, named PA-Cre 3.0, suggesting the original PA-Cre has leaky effect due to non-cleaved form from P2A sequence. Also, they optimized each sequence of Magnet pair for preventing the transgenic line from unexpected recombination in their genomes. However, **it is not clear** why the sequence optimization makes PA-Cre 3.0 free from non-cleaved form. The putative model, traffic jam, author suggested, would not be the reason. If this hypothesis is right, the author should show explainable data.

3. However, suspiciously, CAG promoter-driven Magnet-opti showed non-cleaved form band around 75kDa. The authors mentioned non-cleaved form is a possible reason why original PA-Cre has basal effect. Also, they suggested putative hypothesis the influence of non-cleaved form for basal effect. I mean, because as not like as other optogenetic tools, basal effect of Cre and Flp recombinase activity is critical. However, a non-cleaved form still exists in CAG promoter-driven PA-Cre. Even CAG promoter has somehow strong driving force compared to CMV, transient transfection in HEK293T cell does not fulfill the CMV vs CAG. So, the author should correct this issue with major concern #2.

We really appreciate this reviewer for providing important concerns #1 and #3 regarding the role of “non-cleaved form” in dark leak activities of PA-Cre. To address this concern and clarify the relationship between codon modification and leak activity in dark, we have measured the expressions of PA-Cre mRNA, protein and also Cre-loxP recombination efficiency using qPCR, Western blotting and luciferase reporter assay in this revision, respectively.

First, in this revision, we constructed HA-tagged Magnets and Magnets-opti (i.e. PA-Cre 3.0) constructs to examine the effect of codon modification on PA-Cre expression and function (Fig. 3a,b/R1a,b). The anti-Cre antibody, which we used in this manuscript, has been discontinued and we were about to run out of this reagent (Abcam Cat# ab24608). Alternatively, we tested several anti-Cre antibodies provided by different vendors (Cell Signaling, Cat# 15036, BioLegend Cat# 900901). However, none of them could detect Cre proteins fused with Magnets. Therefore, in this revision, we have used and characterized HA-tagged versions of PA-Cre constructs in order to quantify the protein expressions. A positive outcome using HA antibody was that there were no more non-specific bands observed in the cell lysates, allowing us to quantify the protein expressions of PA-Cre constructs more precisely (please compare “IB: HA” in Fig. 3d & S6a,c to “IB: Cre” in Fig. 2b & S2c).

Second, we transfected these constructs into HEK 293T cells and measured their mRNA expression 48 hours after transfection. The result of qPCR demonstrates that the mRNA expression of Magnets-opti (i.e. PA-Cre 3.0) was significantly decreased to approximately 40-50% compared with Magnets (Fig. 3c/R1c). This is consistent with the phenomenon that codon modification alters

mRNA expression, which has been reported recently (Fu *et al.* J. Biol. Chem. 2018 / Wu Q, *et al.* *eLife*, 2019). Then, to examine whether PA-Cre transcription level alters the protein expression and leak activity in dark, we titrated transfection amount to adjust the mRNA/protein expression levels in Magnets and Magnets-opti (Fig. 3d/S6b,c). Both Magnets and Magnets-opti protein expressions were gradually increased along with the increase of PA-Cre mRNA expressions (Fig. S6c,d). The expression of non-cleaved form in both constructs also increased, being correlated with total protein level (Fig. S6c,e).

Next, we examined the effect of PA-Cre transcript/protein level on Cre-*loxP* recombination using double-floxed reporter and luciferase assay (Fig. 1a). The dark leak of Magnets was increased gradually, as being correlated with the increase of PA-Cre mRNA and protein expressions (Fig. S6f). On the other hand, the dark leak of Magnets-opti (PA-Cre 3.0) did not increase even while the transcripts and proteins increased (Fig. S6f). Because of low dark leak, the fold-induction was ~200-fold in Magnets-opti groups (transfection: 75% and 100%) while, at low protein expression conditions (transfection: 17.5%, 30% and 40%), Magnets also showed ~200-fold induction with light (Fig. S6f). These results suggest that 1) the primary reason of high dark leak in Magnets-based PA-Cre (1.0) is excess expressions of the mRNA and proteins, not the non-cleaved form; 2) moderate mRNA and protein expression level in Magnets-opti could prevent dark leak. We believe that excess mRNA and protein expressions of PA-Cre constructs might cause spontaneous dimerization of nMag-pMag and/or CreN59-CreC60, resulting in leak activity in dark. We really appreciate this reviewer's helpful comments for us to experimentally address the mechanism underlying lower dark leak in Magnets-opti, PA-Cre 3.0.

In addition, we found that Magnets-opti, PA-Cre 3.0, has a unique advantage of minimizing the dark leak because dark leak of Magnets-opti (transfection, 100%) is significantly lower than Magnets (transfection, 40%) while there is no significant difference in the protein amount of cleaved and non-cleaved forms between Magnets-opti and Magnets (Fig. 3d-g/R2a-d). In particular, dark leak in Magnets-opti did not increase even while mRNA and protein increased (transfection: 75% and 100%)(Fig. S6f). According to a report (Kimchi-Sarfaty *et al.* *Science*. 2007), codons used in Magnets might affect protein folding and increase spontaneous nMag-pMag/CreN59-CreC60 associations, which can induce dark leak. Our research findings will be valuable to let technology developers know that codon modification is crucial for developing and optimizing genetically encoded tools. We added this to discussion in this revised manuscript.

Fig. R2 Examining the effect of codon modification on PA-Cre function. (a) Representative Western blotting images of Magnets- and Magnets-opti-transfected HEK 293T cells. CAG-driven HA-tagged PA-Cre proteins were detected by using anti-HA antibody. Loading schematics showed transfection DNA amounts of PA-Cre plasmids. Empty vector was used to adjust total amount of transfection for the cells. Protein expression of Magnets was adjusted to Magnets-opti expression, modifying Magnets plasmid DNA amount. Top bands (~75kDa), non-cleaved form of PA-Cre; bottom bands, cleaved carboxyl-terminal portion of PA-Cre construct (NLS-pMag-CreC60-HA). β -tubulin was examined as an internal control in the cells. Quantification of protein expression of cleaved component of PA-Cre, NLS-pMag-CreC60-HA (b) and non-cleaved form (c, n.s. not significant, Magnets 40 v.s. Magnets-opti 100; *t*-test, *n*=6, mean \pm s.d.). (d) Comparison of PA-Cre activities between Magnets and Magnets-opti using the same protein expression condition. Luc assays were conducted with double-floxed inverted Fluc reporter in HEK 293T cells. Transfected cells were kept in the dark or under blue light illumination (blue LED, 447.5nm, 8.28W/m², repeated 20sec light and 60sec dark for 12 h). The black numbers show fold-induction value (n.s. not significant, *****P*<0.0001; one-way ANOVA with multiple comparison, *n*=8, mean \pm s.d.).

2. In figure 1f, **it is not clear** to see the anti-Cre band in lane 3 (CMV promoter-driven Magnets-opti). Supposing not expressed, no non-cleaved form should be detected and also no basal effect in the dark state. It is also linked with figure 1g in that how CMV promoter-driven Magnets-opti with blue light illumination induced luciferase activity. The author, I think, should reconsider this issue.

We thank this reviewer for this valuable comment. To address this concern, we compared the protein expression of CMV and CAG promoter-driven Magnets-opti constructs in this revision. As we mentioned in major concerns #1, the anti-Cre antibody was discontinued by the vendor, limiting its use. Also, the non-specific band around 50kDa overlapped with the PA-Cre cleaved form (NLS-pMag-CreC60, Fig. 2b). Therefore, we first constructed and then used HA-tagged Magnets and Magnets-opti (both CMV and CAG promoter-driven) to detect the protein expressions of PA-Cre constructs using anti-HA blotting. In this revised approach, we could clearly detect the proteins in Western blotting, allowing us to confirm that that CMV-driven Magnets-opti proteins were expressed in the cells (Fig. S6a/R3).

Fig. R3 Characterization of CMV promoter-driven Magnets-opti 2A and IRES constructs. (a) HA-tagged PA-Cre construct map. (b) qPCR for the construct listed. One-way ANOVA with multiple comparisons (**** $P < 0.0001$). (c) Representative Western blotting images of HA and β -tubulin in HEK 293T cells expressing PA-Cre constructs.

4. In supplementary figure 5., figure 2e expression marker would be needed for image analysis (i.e. EGFP for PA-Cre and Flp-infected) to count PA-Cre expressed cells. In example, it is not convinced Magnets-opti-IRES vector was expressed or not.

We appreciate this reviewer for this helpful comment. Following this concern, we conducted expression profiling of CMV-driven Magnets-opti-IRES (Fig. R3). In the result, we detected mRNA expression of Magnets-opti-IRES, which is comparable to CMV-driven Magnets-opti. However, the protein expression of Magnets-opti-IRES construct was too low to detect. The results suggest that codon modification and IRES system turned the protein expression down in CMV-driven Magnets-opti-IRES while CAG-driven Magnets-opti-IRES responded to blue light (Fig. 2c). We have added the new results of IRES-related constructs to this revised manuscript (Fig. 2c and S4).

5. **The authors should explain in figure 2a-c more detail and precisely.** For example, it is not matched in that if the author had not used Tet-off system (No Dox+ in the illustration and figure legends at all!), the mCherry positive cells would have existed almost all cells in the target regions. So, it is not clear, and I cannot give comments in figure 2a-c.

We appreciate this reviewer for this helpful comment to improve our manuscript. Also, we apologize the previous figure and legend were not clear and sufficient for getting this reviewer convinced with our AAV8-RAM-Tet-off-PA-Cre 3.0 system. We have added more details in this section. Also, in this revision, we conducted additional AAV8-infected mouse experiments using Dox-fed mouse cohorts to provide another negative control group as well (Fig. S7c/R4). The new results

Fig. R4 Testing Dox on AAV8-RAM-Tet-off-PA-Cre 3.0 in vivo mouse brains. Representative immunofluorescent images of red fluorescent protein, mCherry (red) and neuronal activity marker, c-fos (green) and Draq5 (blue, nuclear staining) in amygdala (dashed line) infected with AAV8-RAM-Tet-off-PA-Cre 3.0 and AAV8-hSyn-DIO-mCherry viruses in doxycycline diet-fed (+ Doxycycline) and regular diet (- Doxycycline) group in the conditions of BL illumination.

in vivo are consistent with *in vitro* results using mouse primary neuronal cells (Fig. S7a-b), revealing that AAV8-RAM-Tet-off-PA-Cre 3.0 is functional *in vivo* mouse brains.

6. In supplementary figure 5, the authors insist higher promoter activity minimizes the leak of PA-Cre in the dark state. However, in supplementary figure 11, the authors mention excess expression possibly results spontaneous complementation of PA-Cre with mentioning in the discussion part “Particularly, excess expression weaker promoter may induce lower recombination issue.” (line 196-201). The authors should clear this issue.

We would like to apologize to the reviewer for this confusion. Following the above concerns #1 & 3, we investigated the mechanisms by which codon modification improves the dark leak. We have explained the detailed information in response to #1 & 3 comments and added new results to Fig. 3 and Fig. S6. As a result of mRNA/protein measurements and Cre-loxP recombination analysis, we concluded that excess protein expression of PA-Cre construct is a primary cause of dark leak.

6-1 The author also verified the expression level of “CMV vs CAG” when it comes to their tool. There might be no significant level, or CAG would be little more powerful than CMV from cell types to cell types (Qin et al., PLOS ONE 2010).

We thank this reviewer for this helpful comment. Following this question, we newly constructed CMV-Magnets-HA, CMV-Magnets-opti-HA, CAG-Magnets-HA and CAG-Magnets-opti-HA plasmids in order to quantify PA-Cre proteins as explained in the above #1 & 3. The new Western blotting results demonstrate that the expression level of Magnets was almost same between CMV and CAG promoters (Fig. S6a/R5b). On the other hand, the CAG promoter significantly increased the protein expression of Magnets-opti. We really appreciate this reviewer’s comment to examine the effect of promoter types on protein expressions of PA-Cre constructs.

Minor concerns are :

1. In figure 1a, correct ‘codom’ to ‘codon’ under the construct schematic illustration.

We really appreciate this reviewer’s pointing out our error in the figure. Following this advice, we corrected this typo (Fig. 1a).

2. If the non-cleaved form is the major concern for basal effect of original PA-Cre, although I’m concerning the benefit of one vector system, why don’t the authors test two viral vectors with viral titer optimization rather than 2A ribosomal skipping system for fortifying their hypothesis? I think if the leak problem results in excess expression, adjusting PA-Cre expression would be one of the possible optimization.

We appreciate this valuable suggestion that a two-viral vector system might reduce leak activity. However, our goal of this study is to generate and validate PA-Cre 3.0 knock-in mouse resources using single targeting allele. A flox-based approach will be more useful and convenient for further applications, rather than a viral-based approach. However, in agreement with this reviewer’s comment, we believe that

two viral vectors can be still applicable *in vitro* live cells and *in vivo* transient study. We really appreciate to this comment.

3. Correct the figure name for Supplementary Figure 2 and 4.

We apologize for this error and really thank this reviewer for finding it. Following this comment, we corrected figure names.

4. In Figure 1h and Supplementary Figure 5-6, I wonder it is OK for the liver condition of mice after the 16hr light stimulation. 20mW/mm² is extreme light intensity for mice causing heating effect.

We thank this reviewer for the concerns. As benchmark experiments, we followed the previous PA-Cre paper method (Kawano et al. *Nat. Chem. Bio.* 2016) for light stimulation in order to compare PA-Cre 1.0 and 3.0.

5. Describe ‘Restraint’ methods.

Following this comment, we added detailed experimental protocols in the method section. Please see P23. “Adeno-associated viral infection” We appreciate this reviewer’s comment to improve our manuscript.

6. The authors should explain the process of ‘de-optimization’ with detail. Does it mean sequence optimization for PA-Cre3.0 caused protein level decreased?

We apologize for the lack of clarity: ‘de-optimization’ sounds confusing. We do not use this any more in this revised manuscript. To prevent potential DNA recombination of nMag and pMag components in the mouse genome after gene targeting and mouse breeding, we have changed the nucleotide sequence of CreN59-nMag component using NheI/BamHI sites and IDT program for PA-Cre expression in mice (Fig. S3). As a result of this codon modification, the expression of both PA-Cre mRNA and protein significantly decreased compared to the former version (Magnets, “DNA 2.0” program was used to obtain the highest protein expression of PA-Cre construct reported in our previous study: Kawano *et. al.*) (Fig. 2b and Fig. 3). Because PA-Cre expression was reduced due to the codon changes, we had called it ‘de-optimization’ in the previous manuscript. However, its function has been improved, ‘optimized’, as a controllable Cre recombinase while the amino acid sequences are identical. Therefore, we call this new version Magnets-opti. Again, we agree that ‘de-optimization’ sounds confusing. We do not use this any more in this revised manuscript. We really appreciate this reviewer’s advice.

7. The lentiviral expression would also need expression marker like EGFP.

We thank this reviewer for such an important comment. We regard the best way for evaluation of the virus expression is fusing PA-Cre with expression markers such as fluorescence protein. To accomplish this approach, we inserted mKate2 red fluorescence protein into CMV-driven Magnets-opti construct using BamHI or XhoI site (Fig. R6a). However, because the fusion constructs reduced recombination efficiency (Fig. R6b), these constructs were not so useful. Therefore, alternatively, to measure the

Fig. R6 RFP-fused PA-Cre 3.0. (a) RFP-fused PA-Cre construct map. (b) Luc assays of RFP-fused PA-Cre constructs compared to non-fused PA-Cre 3.0. * P<0.05; ** P<0.01 using Student’s t-test (dark vs BL, n.s. not significant).

infection efficiency, we produced another lentivirus, which expresses yellow fluorescence protein (YFP) and then infected into PA-Cre 3.0 B4 mouse embryonic fibroblast cells. As a result of the infection, was an infection efficiency of approximately 12-13% in each condition and there was no statistically difference between each condition (PA-Cre(+) vs (-) / blue light vs dark)(Fig. S8). Also, the single operator has handled lentivirus production and infection at all the times. The quality of viral vectors and transfection efficiency of lentiviral components into HEK 293T cells were also always monitored during all of our lentiviral production and concentration. Therefore, the virus quality and infection efficiency should be consistent in the viral experiment series. We thank this reviewer for this important comment to ensure experimental rigor in our study.

Reviewer #2 (Remarks to the Author):

The manuscript by Morikawa et al. describes the development and in vivo applications of the new version of photoactivatable Cre recombinase (named PA-Cre 3.0). As its previous version PA-Cre, PA-Cre 3.0 is based on the blue light-dependent dimerization system, Magnets. PA-Cre 3.0 has very low recombination activity in dark but high activity when illuminated with blue light which makes PA-Cre 3.0 quite useful in a number of genome engineering applications.

There are two major differences between the PA-Cre and the PA-Cre 3.0 systems: (1) in PA-Cre 3.0, the codons of the nMag gene are 'de-optimized', and (2) PA-Cre 3.0 performs best when relatively weaker promoters (compared to the full blown CMV) are used to express the system (the CAG promoter and the neuron specific RAM promoter were tested). Taken together, these differences, the authors suggest, reduce the 'traffic jam' of ribosomes on the PA-Cre 3.0 mRNA which leads to the more efficient production of the components of the PA-Cre 3.0 system.

The authors successfully tested PA-Cre 3.0 in two in vivo applications: in mouse liver and in neurons in the newly generated mouse line expressing PA-Cre 3.0.

Although the PA-Cre 3.0 system appears to be very useful, it seems far from being fully characterized since its activity depends on the cell type and the reporter system used to measure its activity.

The manuscript is clearly written and easy to follow. The conclusions are supported by the data. The developed PA-Cre 3.0 system will be quite useful in biomedical research.

The following are relatively minor deficiencies that need to be addressed:

The Abstract is too short (that is, not fully informative) for my taste since it does not clarify (even in general terms) what the authors mean by saying "the new version ... performs better than the previously reported ones". It would be nice to have a sentence added so the readers can understand just from the Abstract (without reading the Introduction) why the PA-Cre 3.0 system is considered better.

We thank this reviewer for the helpful comment. According to this reviewer advice, we re-wrote the abstract to be more informative for all readers. Please see Abstract part.

Fig. 1A. There seems to be a mistake in the representation of the double floxed reporter: the order of loxPs and lox2722s is incorrect (in the relative location shown there will be no deletion of the corresponding lox sites).

We really appreciate the reviewer for pointing out this error. We corrected it in the revised figure (Fig. 1a). We thank this reviewer.

It is not completely clear why the authors did not experiment with the IRES-Magnets-opti constructs and the CAG promoter. On one hand, this construct did not work with the CMV promoter but on the other hand, since the nature/strength of the promoter does seem to make a difference for the functional performance of the system, it is not that apparent that the IRES-Magnets-opti construct paired with the CAG promoter will generate the same result.

We appreciate this comment and agree that a CAG promoter may improve the induction of Magnets-opti-IRES2. To test this hypothesis, we constructed a CAG promoter-driven Magnets-opti-IRES during this revision and tested the performance with two different reporters (double-floxed inverted Fluc and Floxed-mCherry) in HEK 293T cells. Please see the new results of luciferase assay (Fig. 2c) and fluorescence expression analysis (Fig. S4a-d). As this reviewer mentioned, "CAG-Magnets-opti-IRES" improved the induction (3.1-fold induction) in luciferase assay compared to "CMV-Magnets-opti-IRES" (1.0-fold induction). However, the induction level is still quite low compared to "CAG-Magnets-opti (PA-Cre 3.0)" (382.3-fold induction). Therefore, we conclude that CAG promoter-driven Magnets-opti is the best construct to develop the various applications including PA-Cre mouse generation.

REVIEWERS' COMMENTS:

Reviewer #1 (Remarks to the Author):

The authors fulfilled the major concerns by suggesting the mechanism of PA-Cre 3.0. Also, the authors suggested a bunch of examples for applying in vivo experiments. Here, I support the opinion that it is worth publishing in Nature Communications under the promise to elucidate some minor concerns.

Here are the minor concerns and discussion:

1. The order of the sentence, "In addition, we confirmed that doxycycline addition ... via the Tet off system.", seems like in awkward. The description of Dox experiment may be followed by right after.
2. In the Online Methods Plasmid construction section, correct the name pRAM-d2TTA-TRE-MCS-WPEW-pA. (WPRE right?)
3. The thing which I concerned about the mechanism why PA-Cre 3.0 with codon modification had resulted in less leak PA-Cre system has been well established. The author suggested experimental output with remedying their hypothesis. However, I think it might need a bridge for saying the whole results have come from the excessive expression of the protein. With the optimistic expected results, I wonder if global translation inhibition (e.g. treatment of CHX or ANI) or DD domain linking to PA-Cre system would helpful. If works well, the mechanisms the authors suggested would be supported. If too low efficiency for recombination, PA-Cre 3.0 has benefic compared to degradation strategies for making mouse line.
4. In Figure 6., the authors should show PA-Cre level (Cre has been fully disappeared after the RFP expression). For concrete authors' purpose for bar the mice or cells from the unexpected conditions from continuous Cre activity, it will be needed. (Last but worth, then how can we free from the continuous Flp activity?)

Reviewer #2 (Remarks to the Author):

I am satisfied with the response and the changes made.

Reviewer #1 (Remarks to the Author):

The authors fulfilled the major concerns by suggesting the mechanism of PA-Cre 3.0. Also, the authors suggested a bunch of examples for applying in vivo experiments. Here, I support the opinion that it is worth publishing in Nature Communications under the promise to elucidate some minor concerns. Here are the minor concerns and discussion:

1. The order of the sentence, “In addition, we confirmed that doxycycline addition ... via the Tet off system.”, seems like awkward. The description of Dox experiment may be followed by right after.

We really thank this reviewer for pointing out this. Following this advice, we revised the sentences (highlighted).

2. In the Online Methods Plasmid construction section, correct the name pRAM-d2TTA-TRE-MCS-WPEW-pA. (WPRE right?)

We really appreciate this reviewer’s finding our typo in the method. Following this, we corrected this typo.

3. The thing which I concerned about the mechanism why PA-Cre 3.0 with codon modification had resulted in less leak PA-Cre system has been well established. The author suggested experimental output with remedying their hypothesis. However, I think it might need a bridge for saying the whole results have come from the excessive expression of the protein. With the optimistic expected results, I wonder if global translation inhibition (e.g. treatment of CHX or ANI) or DD domain linking to PA-Cre system would helpful. If works well, the mechanisms the authors suggested would be supported. If too low efficiency for recombination, PA-Cre 3.0 has benefic compared to degradation strategies for making mouse line.

We really appreciate the helpful advice provided by this reviewer. First, in this revision following this advice, we prepared a new PA-Cre construct using BamHI sites of Magnets (original), CMV promoter (i.e. pcDNA 3.0 vector) and “destabilization domain” (DD,

KLSHGFPPEVEEQDDGTLPMSCAQESGMDRHPAACASARINV), which has been used for destabilizing eGFP known as “d2eGFP” (Warren et al. *Cell Stem Cell* 2010 Sep 29). This new construct, PA-Cre (Magnets) linked with DD (“Magnets-ds”, **Fig. RII-1a**), worked better than PA-Cre Magnets. The induction was improved with significant reduction of dark leakiness (**Fig. RII-1b,c**). This result supports the mechanisms we have proposed in which PA-Cre has been improved by codon modification. However, PA-Cre 3.0 (codon-modified, “Magnets-opti”) works better than this new version, Magnets-ds, with lower dark leak (**Fig. RII-1b,c and Fig. S4a**). Also, Magnets-ds may possibly still have pMag-nMag recombination in mouse genome even if its mouse line is generated because of high homology between pMag-nMag.

Therefore, the results reveal that PA-Cre 3.0 (Magnets-opti and CAG promoter) is the best and most ideal for mouse targeting and further applications *in vivo*.

In addition, following this reviewer's advice, we examined the effect of anisomycin (ANI), a translation inhibitor, on dark leak of PA-Cre Magnets in mammalian cell line, HEK293T cells (Fig. RII-2a). While ANI (1,000-5,000nM) could significantly reduce dark leak of PA-Cre Magnets in the cells (Fig. RII-2b), likely compatible with PA-Cre 3.0 (dark), the cell viability and translation were obviously affected, showing lower cell density (Fig. RII-2c). Higher concentrations of ANI induced further unhealthy conditions of treated cells. YFP transient overexpression was used to examine the effect of ANI on global translation in the live cells. The results reveal that the new results obtained from the revised experiments using ANI were not clearly conclusive because of unhealthy cell conditions.

Also, we examined the effect of cycloheximide (CHX), another global translational inhibitor, on dark leak of PA-Cre Magnets, following this reviewer's helpful advice. While CHX could also reduce dark leak of PA-Cre Magnets, as we observed in ANI treatment, CHX affected cell viability (Fig. RII-3a-c). Therefore, ANI and CHX are not ideal reagents to test our hypothesis while DD construct can successfully confirm that our proposed mechanisms underlying PA-Cre 3.0 improvement with codon modification in this revised manuscript.

Thanks to this reviewer's comments and advices, the molecular mechanism underlying PA-Cre improvement with codon optimization has been clearly confirmed in our revised manuscript. We really appreciate this reviewer's generous help and thoughtful advice to improve our manuscript.

4. In Figure 6., the authors should show PA-Cre level (Cre has been fully disappeared after the RFP expression). For concrete authors' purpose for bar the mice or cells from the unexpected conditions from continuous Cre activity, it will be needed. (Last but worth, then how can we free from the continuous FIp activity?)

We really apologize for the misunderstanding from this reviewer regarding our rationale to generate all-in-one version of PA-Cre 3.0 mouse. Because mouse breeding takes time for crossing three different lines such as PA-Cre 3.0, reporter and target gene flox, we have proposed this all-in-one concept of PA-Cre 3.0 mouse generation. Cre and FIp continuous expressions using cell/tissue-specific promoters have been widely used in mouse genetics and disease modeling. We do not have major concerns on Cre and FIp expressions in majority of target cells of existing mouse models. Following this reviewer's comment, we revised our manuscript regarding Figure 6 (highlighted). We truly appreciate this reviewer's comment to state our rationale of the all-in-one version of PA-Cre 3.0 mouse.